# Learning Linear Attention in Polynomial Time

**Morris Yau**
MIT CSAIL
morrisy@mit.edu

**Ekin Akyürek**
MIT CSAIL
akyurek@mit.edu

**Jiayuan Mao**
MIT CSAIL
jiayuanm@mit.edu

**Joshua B. Tenenbaum**
MIT Brain and Cognitive Sciences
jbt@mit.edu

**Stefanie Jegelka**
TUM Munich, MCML, MIT CSAIL
stefje@mit.edu

**Jacob Andreas**
MIT CSAIL
jda@mit.edu

## Abstract

Previous research has explored the expressivity of Transformer models in simulating Boolean circuits or Turing machines. However, the efficient learnability of Transformers from data has remained an open question. Our study addresses this gap by providing the first polynomial-time learnability results (specifically strong, agnostic PAC learning) for single-layer Transformers with linear attention. We show that learning the optimal multi head linear attention can be recast as finding the optimal kernel predictor in a suitably defined RKHS. Moving to generalization, we construct an algorithm that, given a dataset, checks in polynomial time whether the set of best fit multi head linear attention networks on this data all perform an identical computation–a powerful notion for out of distribution generalization. We empirically validate our theoretical findings on several canonical tasks: learning random linear attention networks, key–value associations, and learning to execute finite automata. Our findings bridge a critical gap between theoretical expressivity and learnability of Transformer models.

## 1 Introduction

Transformers are the dominant neural architecture used in language modeling. A growing body of work seeks to explain the behavior of trained Transformers and characterize their learnability [Pérez et al., 2019, Edelman et al., 2022b, Hahn, 2020, Merrill and Sabharwal, 2023, Merrill et al., 2022, 2021, Liu et al., 2022, Feng et al., 2023, Edelman et al., 2022a, Wei et al., 2021, Zhang et al., 2024, Trauger and Tewari, 2023, Chen and Li, 2024]. While a large body of work shows that Transformers are *expressive* enough to implement important models of computation, it remains an open question whether these constructions may be efficiently *learned*. Even verifying that a trained model has successfully learned a computational procedure (uniform circuit family) has remained challenging.

Existing work shows positive results on how Transformer-like architectures can express diverse computations, including simulating universal Turing machines [Li et al., 2024], evaluating sentences of first-order logic [Barceló et al., 2020], and recognizing various formal languages [Strobl et al., 2024]. On the other hand, results on learnability in polynomial time and samples with provable guarantees tend to rely on strong data-generating assumptions, e.g., Gaussian data, etc. [Zhang et al., 2023, Jelassi et al., 2022, Tian et al., 2023, Oymak et al., 2023, Fu et al., 2023, Tarzanagh et al., 2024, Deora et al., 2023]. This brings us to our first motivating question.

*Is there an efficient algorithm in time and samples that learns the optimal parameters of a class of Transformer models for any dataset?*

In this paper, we establish the strong, agnostic PAC-learnability of linear attention. Linear attention variants (kernel, gated, flash, etc.) Yang et al. [2025, 2024], mLSTM in xLSTM Beck et al. [2024],

Retnet Sun et al. [2023], Mamba2 Dao and Gu [2024], DeltaNet Schlag et al. [2021]) have recently matched or outperformed softmax attention in language and vision benchmarks, underscoring the practical value of their theory; Ahn et al., 2024, Katharopoulos et al., 2020). Despite its name, linear attention is not linear and its loss landscape is nonconvex. We focus our analysis on multi-head linear attention networks, or MHLAs for regression tasks. An MHLA is parameterized by two matrices $(V_h, Q_h)$ for each of $H$ heads as such $\Theta = \{(V_h, Q_h)\}_{h \in [H]}$. A one layer MHLA computes $Y = \sum_{h \in [H]} V_h Z (Z^T Q_h Z)$. Here key and query matrices are fused into one, as they multiply one another directly.

We first show that the computation performed by MHLAs can be reformulated as an elementwise product between two larger matrices $\langle W, \mathcal{X}(Z) \rangle$, where $W = \sum_{h \in [H]} \text{flatten}(V_h)\text{flatten}(Q_h)^T$ and $\mathcal{X}(Z)$ is a fixed cubic polynomial function of $Z$. Consequently, optimizing over the class of $H$-head MHLA models is equivalent to optimizing over the class of rank-$H$ matrices $W$. Furthermore, in the full-rank space of $d^2 \times d^2$ matrices, optimization of $W$ can be performed via linear regression with time polynomial in the inverse target error and size of the dataset. Finally, decomposing an optimal $W$ via SVD recovers an MHLA model with no more than $d^2$ heads that is then guaranteed to compete against the best MHLA parameters—establishing our agnostic learning result (the learned model competes against the best choice of parameters in the hypothesis class).

Next, achieving zero training and validation loss does not by itself certify that a model has learned a target computation well enough to generalize out of distribution. Imagine learning arithmetic from input output pairs alone. Many distinct parameter settings can fit the same data, and fail for larger length inputs. We therefore ask:

*Is there a data-dependent, efficiently checkable condition that forces every empirical-risk minimiser to realise the same function?*

For MHLAs the answer is **yes**. Define the second-moment matrix of the cubic feature map $\mathcal{X}$ as

$$\Lambda_D = \mathbb{E}_{(Z,y) \in D}\big[\mathcal{X}(Z)\,\mathcal{X}(Z)^\top\big].$$

If $\Lambda_D$ is full rank—our *certifiable identifiability* criterion—then *all* empirical-risk minimisers of MHLA coincide on every input. The test runs in polynomial time and is unaffected by parameter redundancies such as rescaling $V$ and $Q$.

Combining this certificate with our expressivity result yields a polynomial-time procedure that (i) learns any circuit family implementable by MHLA whenever the training data satisfy the criterion, and (ii) provably recovers, for example, a bounded-history universal Turing machine from its input–output traces (Appendix C). Once learned, the MHLA simulates any such Turing machine on any input within the prescribed size budget.

In the experimental section, we validate our theoretical findings. In Section 4.1, we train multiple models using stochastic gradient descent on a dataset generated by a single linear attention network's output. Our results demonstrate that multi-head linear attention outperforms both single-layer linear attention and multi-layer linear attention, achieving comparable results to our Algorithm 1. In Section 4.2, we show that our proposed certificate directly correlates with generalization error even for models trained using stochastic gradient descent. In summary:

- We provide a polynomial time algorithm that, given any dataset, finds the best fit parameters for multi head linear attention and generalizes with polynomial data, i.e., strong agnostic PAC learning (Section 2.1).

- We find an efficiently checkable condition (certifiable identifiability) on the training dataset that certifies every empirical risk minimizer of a MHLA is functionally equivalent, and therefore has the same behavior out of distribution (Appendix A see Lemma A.3).

- We study empirically the value of overparameterization with multiple heads vs. multiple layers in Section 4.1. We verify our certificates empirically on the associative memory task in Section 4.2.

---

**Algorithm 1** MHLA Learning via Regression

---

1: **Input:** Data $D := \{(Z_i, y_i)\}_{i \in [N]}$ for $Z_i \in \mathbb{R}^{d \times n_i}$ and $y \in \mathbb{R}^d$

2: $\{\mathcal{X}_i\}_{i \in [N]} := \text{ExtractFeature}(D)$, generates

$$\mathcal{X}_i := \begin{bmatrix} \langle z_{1:}, z_{1:} \rangle z_{1n_i} & \langle z_{1:}, z_{2:} \rangle z_{1n_i} & \cdots & \langle z_{1:}, z_{d:} \rangle z_{dn_i} \\ \langle z_{2:}, z_{1:} \rangle z_{1n_i} & \langle z_{2:}, z_{2:} \rangle z_{1n_i} & \cdots & \langle z_{2:}, z_{d:} \rangle z_{dn_i} \\ \vdots & \vdots & \ddots & \vdots \\ \langle z_{d:}, z_{1:} \rangle z_{1n_i} & \langle z_{d:}, z_{2:} \rangle z_{1n_i} & \cdots & \langle z_{d:}, z_{d:} \rangle z_{dn_i} \end{bmatrix}. \tag{1}$$

3: Create dataset $\{X_{i,a}\}_{i \in [N], a \in [d]}$. Let $X_{i,a} \in \mathbb{R}^{d^2 \times d^2}$ be a matrix that is comprised of $\mathcal{X}_i$ in the $a'th$ block of $d$ rows and 0 everywhere else:

4:
$$X_{i,a} = \begin{bmatrix} 0 & \ldots & \mathcal{X}_i^T & \ldots & 0 \end{bmatrix}^T \tag{2}$$

5: Let $\hat{W} \in \mathbb{R}^{d^2 \times d^2}$ be regressor:

$$\hat{W} := \underset{W \in \mathbb{R}^{d^2 \times d^2}}{\arg\min} \sum_{i \in [N]} \sum_{a \in [d]} \left( \langle W, X_{i,a} \rangle - y_{i,a} \right)^2 \tag{3}$$

where $y_{i,a}$ is the $a$'th coordinate of $y_i$.

6: Take the SVD of $\hat{W} = AB^T = \sum_{i \in [\hat{H}]} A_i B_i^T$ where $\hat{H}$ is the rank of $\hat{W}$.

7: $V_h = \text{Fold}(A_h)$ and $Q_h = \text{Fold}(B_h)$ where $\text{Fold} : \mathbb{R}^{d^2} \to \mathbb{R}^{d \times d}$ takes a vector $p := [p_{ij} \text{ for } i \in [d] \text{ and } j \in [d]]$ and reshapes into a matrix $P \in \mathbb{R}^{d \times d}$ such that $P_{ij} = p_{ij}$.

8: **Return:** $\{V_h, Q_h\}_{h \in [\hat{H}]}$

---

## 2 Technical Overview

We start with basic definitions of a multi-head linear attention (MHLA) module, an attention module without the softmax activation.

**Definition 2.1** (Multi-Head Linear Attention). Let $Z \in \mathbb{R}^{d \times n}$ be a matrix of input data. Let $\Theta = \{(V_h, Q_h)\}_{h \in [H]}$ be a set of parameters where each $V_h, Q_h \in \mathbb{R}^{d \times d}$ denotes value and key-query matrices for all heads $h \in [H]$. We say $\Theta \in \Omega_H$ where $\Omega_H$ is the space of sets of $H$ ordered tuples of $d \times d$ matrices. We define *multi-head linear attention (MHLA)* to be the function $\text{MHLA}_\Theta : \mathbb{R}^{d \times n} \to \mathbb{R}^{d \times n}$,

$$\hat{Y} = \text{MHLA}_\Theta(Z) = \sum_{h \in [H]} V_h Z(Z^T Q_h Z), \tag{4}$$

where $\hat{Y} \in \mathbb{R}^{d \times n}$ is the output of the one layer linear attention. We will primarily be interested in the rightmost column vector output by $\text{MHLA}_\Theta$ (e.g., as in auto-regressive language models), which is:

$$\hat{y} = \text{MHLA}_\Theta(Z) = \sum_{h \in [H]} V_h Z(Z^T Q_h Z[:, n]), \tag{5}$$

where $Z[:, n]$ is the $n$'th column of $Z$.

### 2.1 Polynomial-time learnability

Our main result is that MHLA is learnable in polynomial time. Colloquially, Algorithm 1 returns an MHLA that attains the global minimum of the training loss and requires as few as $\text{poly}(d, \epsilon^{-1}, \log(\delta^{-1}))$ samples to achieve $\epsilon$ generalization error with probability $1 - \delta$. Our algorithmic guarantees do not require the data to be "realizable" (that is, the data need not be generated by an underlying MHLA).

**Theorem 2.2** (Learnability of Linear Attention). *Let $D$ be a dataset $D = \{Z_i, y_i\}_{i \in [N]}$ drawn i.i.d. from a distribution $\mathcal{D}$ where each $Z_i \in \mathbb{R}^{d \times n_i}$, $y_i \in \mathbb{R}^d$. The embedding dimension $d$ is fixed across the dataset, whereas $n_i$ can be different for each datapoint. Let $n_{max} = \max_{i \in [N]} n_i$ be the maximum sequence length, and let $\Omega_H$ be the space of $H$ pairs of value and key-query matrices $\{(V_h, Q_h)\}_{h \in [H]}$ for any $H \in [1, \infty)$. Then there is an algorithm (Algorithm 1) that runs in time $O(Nd^4 n_{max} \epsilon^{-1})$ and that, given input–output pairs $\{(Z_i, y_i)\}_{i \in [N]}$, returns $\hat{\Theta} = \{(\hat{V}_h, \hat{Q}_h)\}_{h \in [\hat{H}]} \in \Omega_{\hat{H}}$ for $\hat{H} \leq d^2$*

*such that with probability $1 - \delta$,*

$$\mathbb{E}_{(Z,y)\in\mathcal{D}}\left[\|MHLA_{\hat{\Theta}}(Z) - y\|^2\right]$$

$$- \min_{\Theta\in\Omega_H} \mathbb{E}_{(Z,y)\in\mathcal{D}}\left[\|MHLA_{\Theta}(Z) - y\|^2\right] \leq \epsilon \quad (6)$$

*with sample complexity $N = O\left(\frac{1}{\epsilon}\left(d^4 + \log(\delta^{-1})\right)\right)$.*

Below we describe the high-level ideas behind the algorithm; a formal proof is given in Appendix D. Note that if we are purely concerned with guaranteeing that we can find a global minimum of the training loss, we may remove the i.i.d. assumption: Algorithm 1 is always within error $\epsilon$ of the optimal training loss. This is also detailed in Appendix D. Specific issues related to generalization over autoregressive sequences rather than i.i.d. data are handled in the UTM learning result with a standard union bound on the sample complexity; see Section F.2.

The main idea behind Algorithm 1 is to construct a feature mapping $\mathcal{X} : \mathbb{R}^{d\times n} \to \mathbb{R}^{d\times d^2}$ from the data covariates $Z$ with entries $z_{ij}$ for the entry in the $i$'th row and $j$'th column and rows $z_{1:}, z_{2:}, ..., z_{d:} \in \mathbb{R}^n$ to a feature space of dimension $d \times d^2$. The map $\mathcal{X}(Z)$ is defined as:

$$\mathcal{X}(Z) :=$$

$$\begin{bmatrix} \langle z_{1:}, z_{1:}\rangle z_{1n} & \langle z_{1:}, z_{2:}\rangle z_{1n} & \cdots & \langle z_{1:}, z_{d:}\rangle z_{dn} \\ \langle z_{2:}, z_{1:}\rangle z_{1n} & \langle z_{2:}, z_{2:}\rangle z_{1n} & \cdots & \langle z_{2:}, z_{d:}\rangle z_{dn} \\ \vdots & \vdots & \ddots & \vdots \\ \langle z_{d:}, z_{1:}\rangle z_{1n} & \langle z_{d:}, z_{2:}\rangle z_{1n} & \cdots & \langle z_{d:}, z_{d:}\rangle z_{dn} \end{bmatrix}. \quad (7)$$

Here, we index the rows of $\mathcal{X}(Z)$ by $j \in [d]$ and the columns by all tuples $(k, \ell) \in [d]^2$ such that $\mathcal{X}(Z)_{j,(k,\ell)} = \langle z_{j:}, z_{k:}\rangle z_{\ell n}$. At a high level, Algorithm 1 is a kernel method defined by the feature mapping $\mathcal{X}$. The learned kernel predictor (a regressor) can be mapped back onto a set of parameters $\{\hat{V}_h, \hat{Q}_h\}_{h\in\hat{H}}$ for an MHLA with no more than $d^2$ heads via SVD. Hence, the relaxation translates into more heads. Interestingly, in our experiments in Section 4.1, $d^2$ heads also benefit learning with SGD.

**Proof Idea:** Much of the notation in this section is defined in Algorithm 1. First we write down the loss, and observe that a one-layer attention network is a quadratic polynomial in $\{V_h, Q_h\}_{h\in[H]}$ with input features $X_{i,a}$:

$$\mathcal{L}_{\Theta}(\{(Z_i, y_i)\}_{i\in[N]}) = \frac{1}{N}\sum_{i\in[N]}\sum_{a\in[d]}(\langle\mathcal{T}_{\Theta}, X_{i,a}\rangle - y_{i,a})^2 \quad (8)$$

with

$$\mathcal{T}_{\Theta} := \sum_{h\in[H]} \text{flatten}(V_h)\text{flatten}(Q_h)^T$$

$$= \sum_{h\in[H]} \begin{bmatrix} V_{h,00}Q_{h,00} & V_{h,00}Q_{h,01} & \cdots & V_{h,00}Q_{h,dd} \\ V_{h,01}Q_{h,00} & V_{h,01}Q_{h,01} & \cdots & V_{h,01}Q_{h,dd} \\ \vdots & \vdots & \vdots & \\ V_{h,dd}Q_{h,00} & V_{h,dd}Q_{h,01} & \cdots & V_{h,dd}Q_{h,dd} \end{bmatrix}$$

Now we relax this objective by replacing $\mathcal{T}_{\Theta}$ with an unconstrained matrix $W \in \mathbb{R}^{d^2\times d^2}$. While $\mathcal{T}_{\Theta}$ is a rank-$H$ matrix, we allow $W$ to be a general matrix, so this relaxation is guaranteed to have a smaller loss. Furthermore, the loss can be optimized via ordinary least squares. Finally, if we apply SVD to $W$ we obtain a set of $d^2$ left and right singular vectors scaled by the square root the magnitude of the singular value. Here the scaled left singular vectors correspond to $\hat{V}_h$ and the scaled right singular vectors correspond to $\hat{Q}_h$ for $h \in [\hat{H}]$. Since the rank of $W$ is no greater than $d^2$ the resulting MHLA satisfies $\hat{H} \leq d^2$. The sample complexity follows from classical results in VC theory [Kearns and Vazirani, 1994]. For a full proof see Appendix D.

## 2.2 Identifiability

A direct implication of our algorithmic result is the construction of an efficiently checkable condition on the data that guarantees every empirical risk minimizer in a family of MHLAs computes the same function. Let $\Lambda_D$ be the second moment of a specific mapping $\mathcal{H}(Z)$ of the data, defined in Lemma A.3.

$$\Lambda_D = \mathbb{E}[\mathcal{H}(Z)\,\mathcal{H}(Z)^T] = \frac{1}{N}\sum_{Z\in D}[\mathcal{H}(Z)\,\mathcal{H}(Z)^T]. \tag{9}$$

Then if $\Lambda_D$ is full rank or equivalently its minimum eigenvalue is greater than zero, then it is guaranteed that MHLA is *identifiable with respect to the data*.

**Lemma 2.3** (Certificate of Identifiability—Informal). *Let dataset $D = \{(Z_i, y_i)\}_{i\in[N]}$ be realizable (see Definition A.2) by an $H$-head MHLA for any $H \geq 1$. Let $\mathcal{H}$ be the uniform family of polynomials $\mathcal{H}_n : \mathbb{R}^{d\times n} \to \mathbb{R}^\psi$ for $\psi := \binom{d}{2}d + d^2$ defined as in Algorithm 2. For convenience we drop the subscript of $n$ and write $\mathcal{H}(Z)$ to mean $\mathcal{H}_n(Z)$ for $Z \in \mathbb{R}^{d\times n}$. Finally, define $\Lambda_D \in \mathbb{R}^{\psi\times\psi}$ to be the second moment of the data features:*

$$\Lambda_D := \mathbb{E}_D\left[\mathcal{H}(Z)\mathcal{H}(Z)^T\right]. \tag{10}$$

*Then if the eigenvalue $\lambda_{\min}(\Lambda_D) > 0$, we say that $MHLA_\Theta$ is certifiably identifiable with respect to $D$. That is, for every pair of empirical risk minimizers $\Theta, \Theta' \in \Omega_H$*

$$MHLA_\Theta = MHLA_{\Theta'} \tag{11}$$

*i.e., the two models have the same outputs on all inputs.*

**Corollary 2.4.** *There is a polynomial $p : \Omega_H \to \mathbb{R}^\psi$ such that for any pair of parameters $\Theta, \Theta' \in \Omega_H$ we have $MHLA_\Theta = MHLA_{\Theta'}$ if and only if $p(\Theta) = p(\Theta')$.*

The polynomial $p$ defines the equivalence class of parameters that compute the same function. For a formal statement of Lemma 2.3 see Lemma A.3. For handling of errors for approximate empirical risk minimization see Lemma A.7. Moreover, the certificate given by Algorithm 2 is not the only choice of feature mapping $\mathcal{H}$ that would certify identifiability; Lemma E.1 gives a general certificate for identifiability. One way to interpret Corollary 2.4 is that two MHLA models parameterized by $\Theta$ and $\Theta'$ compute the same function if and only if they are the same linear function in a specific feature space (akin to matching coefficients in polynomial regression), which in turn is true if $p(\Theta) = p(\Theta')$ for the polynomial $p$ given in Corollary A.4. Comparing distance between the coefficients in the range of $p$ is essentially the only meaningful metric of distance that is agnostic to the choice of dataset.

Finally, we answer a few natural questions related to identifiability which we briefly summarize here. Firstly, perfectly noisy input data is identifiable under weak assumptions on the moments of the noise (see Lemma A.5). Secondly, the model class of MHLA with at least $d^2$ heads is certifiably identifiable from the second moment condition alone, and does not require realizability of the data (see Lemma A.6). Finally, we empirically verify the min eigenvalue of $\Lambda_D$ predicts the generalization behavior of SGD for MHLA for the problem of learning key–value memories (see Figure 2).

## 3 Application to learning Universal Turing Machines.

In Appendix B, we demonstrate that MHLAs can (autoregressively) express universal Turing machines with polynomially bounded computation histories. In this context, our identifiability results imply that, given a certifiably identifiable dataset of Turing machines and their computation histories on input words, empirical risk minimization and in particular Algorithm 1 will learn the universal Turing machine in a strong sense (Lemma C.5 for learning, Lemma A.8 with identifiability). That means at test time the learned MHLA will simulate any Turing Machine on any input word up to a given size for a bounded number of steps. For more detail see C

**Lemma 3.1** (Learning UTM from Certifiably Identifiable Data). *Let $D = \{(Z_i, y_i)\}_{i\in[N]}$ be a dataset satisfying $y_i = MHLA_\Theta$ for $\Theta \in \Omega_H$ being the expressibility parameters of Lemma B.1 for the set of TM's/words $(M, x) \in \Delta(\hat{\mathcal{Q}}, \hat{\Sigma}, \hat{n}, \hat{\Phi})$. If $D$ is certifiably identifiable with $\lambda_{min}(\Lambda_D) > \eta$, then there is a $poly(d, N, \hat{Q}, \hat{\Sigma}, \hat{n}, \hat{\Phi}, \eta^{-1})$ time algorithm that outputs a set of parameters $\hat{\Theta} \in \Omega_{d^2}$ such that for all TM's $M$ and input words $x$ in $\Delta(\hat{\mathcal{Q}}, \hat{\Sigma}, \hat{n}, \hat{\Phi})$, we have*

$$CH_{\hat{\Theta}}(M, x)^{c(t)}[:-k_t] = x^t. \tag{12}$$

*The $c(t)$ step of the autoregressive computation history of $\hat{\Theta}$ is equal to the $t$'th step of the computation history of $M$ on $x$.*

## 4 Experiments

In our experiments, we validate our theoretical predictions in settings where Transformers are trained using stochastic gradient descent (SGD), as follows: Firstly, Theorem 2.2 exploits that adding a sufficient number of heads to an MHLA leads to a convex optimization problem after reparameterization. This suggests that over-parameterizing by adding heads may provide optimization benefits. We investigate the role of over-parameterization in multi-head and multi-layer linear attention networks. For random data generated from linear attention networks, we observe that adding more heads achieves faster convergence of training loss than adding more layers. This suggests that while depth is important for expressiveness, the number of heads is important for optimization (Figure 3).

Secondly, we empirically verify the certificate of identifiability provided by Lemma A.3 on datasets for associative memory [Bietti et al., 2023, Cabannes et al., 2024] with different choices of embeddings, demonstrating convergence to the equivalence class of the true parameters when $\lambda_{min}(\Lambda_D) > 0$ and converging to spurious solutions when $\lambda_{min}(\Lambda_D) = 0$ (Figure 2).

### 4.1 Do extra heads help optimization with SGD?

To probe whether more heads facilitate learning in general, we train our convex relaxation and different types of over-parameterized models with SGD on data generated from a single-layer linear attention network. For the data, we initialize a single-layer linear attention network with parameters $V \in \mathbb{R}^{1 \times d}$ and $Q \in \mathbb{R}^{d \times d}$, sampled from a Gaussian distribution $\mathcal{N}(0, \frac{I}{\sqrt{d}})$. Input sequences $Z^i \in \mathbb{R}^{T \times d}$ are sampled from $\mathcal{N}(0, \frac{I}{\sqrt{T}})$, where $i = 1, \ldots, N$, $T = 100$ is the maximum number of time steps, and $N$ is the dataset size. We generate outputs by running the ground-truth network auto-regressively: $y_t^i = V Z_{1:t}^i (Z^i[:, :t] Q Z^i[:, t])$, creating our dataset $\mathcal{D} = \{(Z^i, y^i)\}_{i=1}^N$.

In addition to learning with Algorithm 1, we train three types of models on this data using SGD: (1) multi-head linear attention as in Equation (4); (2) multi-layer linear attention with a single head; (3) an ordinary Transformer network [Vaswani et al., 2017] with softmax attention, multi-layer perceptron blocks, and layer normalization.

Figure 1 illustrates the results. For same experiment with $d = 4$ and $N = 2048$ see Figure 3a in the appendix. Detailed hyperparameters and optimization procedures are described in Appendix G.1.

We observe that multi-head attention scales effectively with an increasing number of heads, resulting in improved performance. Notably, for $d = 2$ or $4$ input dimensions, using $d^2$ heads yields the best performance and is empirically comparable to Algorithm 1, approaching floating-point error precision. Theoretically, $d^2$ is the maximum rank in the relaxation in Algorithm 1. In contrast, multi-layer attention models show diminishing returns and perform worse than single-layer attention. Interestingly, adding more layers can sometimes degrade performance. The full transformer model, which incorporates softmax attention, MLP layers and layer normalization, does not significantly outperform the single-layer linear attention model on this task.

These findings suggest that the type of over-parameterization matters significantly in learning linear attention networks. Interestingly, multi-head architectures appear to be particularly effective—aligned with the structure of Algorithm 1, where the relaxation corresponds to adding more heads.

### 4.2 Does certifiable identifiability predict generalization?

In Lemma A.3, we developed a certificate that provides a sufficient condition for identifiability. To assess the practical relevance of this certificate, we conducted an empirical analysis of convergence in cases where the condition is not satisfied. The results of this analysis are presented in Figure 2.

**Associative Memory** Associative Memory [Bietti et al., 2023, Cabannes et al., 2024] is a task of looking up a value in a table with a query. Via a single head one-layer linear attention model it can be

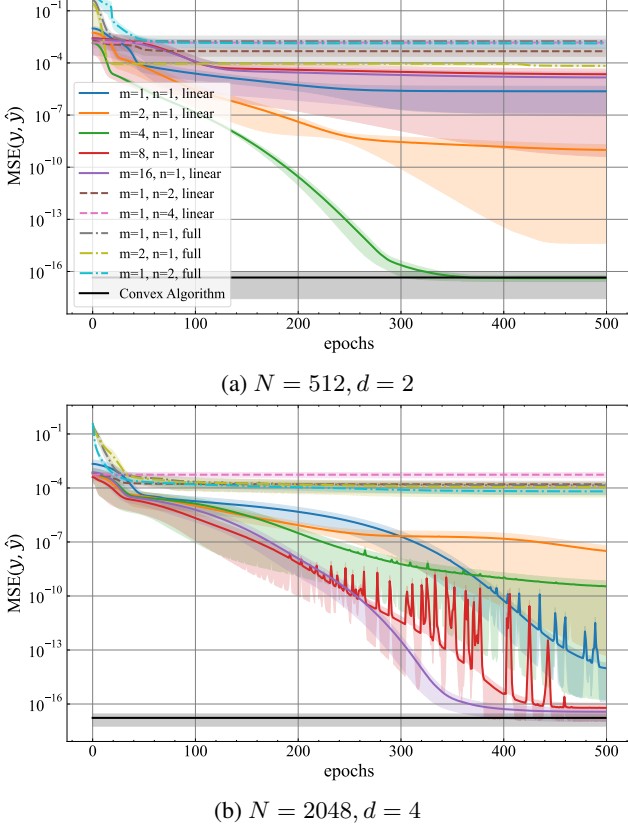

(a) $N = 512, d = 2$

(b) $N = 2048, d = 4$

Figure 1: **Performance comparison of multi-head, multi-layer linear attention models and the original Transformer model (denoted as *full*)**. We trained using SGD on synthetic data generated from a single-layer linear attention model for varying training set sizes ($N$) and input dimensions ($d$), number of heads $m$, and number of layers $n$. Results demonstrate that multi-head architectures converge faster on different input dimensions and match the performance of our algorithm 1 (convex algorithm). Increasing the number of layers or incorporating multilayer perceptrons (MLPs) and layer normalization did not yield consistent improvements. Shading indicates the standard error over three different runs.

represented with ground truth parameters $\Theta = \{V, Q\}$ where $V, Q \in \mathbb{R}^{2d \times 2d}$:

$$V = \begin{bmatrix} 0 & 0 \\ 0 & I_{d \times d} \end{bmatrix} \quad Q = \begin{bmatrix} I_{d \times d} & 0 \\ 0 & 0 \end{bmatrix}.$$

The data $Z$ is drawn as follows: let $k_1, k_2, ..., k_d \in \mathbb{R}^d$ be random variables corresponding to keys in a lookup table, let $v_1, v_2, ..., v_d \in \mathbb{R}^d$ be random variables corresponding to values in a lookup table, let $q \in \mathbb{R}^d$ be a random variable corresponding to a query to the lookup table, and $\zeta \sim \mathcal{N}(0, I)$ be random noise, such that $Z$ and the output vector $y$ are defined as:

$$Z = \begin{bmatrix} k_1 & k_2 & \dots & k_d & q \\ v_1 & v_2 & \dots & v_d & \zeta \end{bmatrix} \tag{13}$$

$$y = \text{MHLA}_\Theta(Z) = \begin{bmatrix} 0 \\ \sum_{j \in [d]} \langle q, k_j \rangle v_j \end{bmatrix}. \tag{14}$$

**Mixture of distributions:** We generate two datasets, one that has identifiable $\lambda_{\min}(\Lambda_D) > 0$ and one that is nonidentifiable with $\lambda_{\min}(\Lambda_D) = 0$. The identifiable dataset is generated with $\{k_j\}_{j \in [d]}$ and $\{v_j\}_{j \in [d]}$ drawn i.i.d $\mathcal{N}(0, I)$. The query $q$ is chosen to be one of the $\{k_j\}_{j \in [d]}$ uniformly at random. The non-identifiable dataset is drawn such that $\{k_j\}_{j \in [d]}$ forms a random unitary matrix,

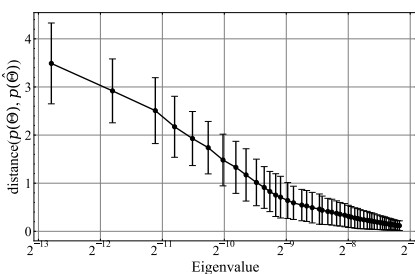
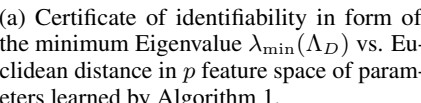
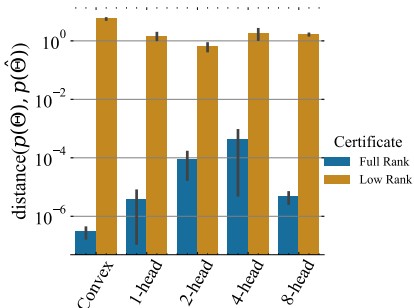

(a) Certificate of identifiability in form of the minimum Eigenvalue $\lambda_{\min}(\Lambda_D)$ vs. Euclidean distance in $p$ feature space of parameters learned by Algorithm 1.

(b) Distance to ground truth parameters in $p$ feature space for certifiably identifiable data (min eigenvalue $= 0.06$) vs. nonidentifiable data (min eigenvalue $= 0$). Here the parameters of MHLA are learned via SGD. Error bars are standard error on three different runs.

Figure 2: **Impact of data distribution on the associative lookup task performance:** We generated training data for an associative lookup task [Bietti et al., 2023, Cabannes et al., 2024] using mixtures of two distributions: (1) Gaussian key and value vectors, and (2) random unitary key and value vectors. By adjusting the mixture probability, we can manipulate the certificate value (minimum eigenvalue of the data covariance matrix), as unitary key–value vectors give rank-deficient "certificates". (a) Algorithm 1: as the minimum eigenvalue increases, Algorithm 1 converges more closely to the true parameters. (b) SGD: SGD learns parameters that are equivalent to the ground truth parameters in $p$ feature space for certifiably identifiable data, but for unidentifiable data, they are far apart in $p$ feature space and therefore compute different functions.

i.e., $\|k_j\| = 1$ for all $j \in [d]$ and $\langle k_j, k_{j'} \rangle = 0$ for all $j \neq j'$. Similarly, $\{v_j\}_{j \in [d]}$ is also drawn from a randomly generated unitary matrix. We draw new random unitary matrices for each datapoint, where $q$ is again chosen to be one of the $\{k_j\}_{j \in [d]}$ uniformly at random. We set $d = 4$ dimensions for both datasets, and draw $N = 2^{14}$ samples for each dataset. We mix the two datasets together with a mixing probability ranging from 95% unidentifiable to 100% unidentifiable. In this manner we generate a spread of datasets with different values for $\lambda_{\min}(\Lambda_D)$ that tend to zero.

**Certifiable Identifiability for Algorithm 1:** For each dataset, we run Algorithm 1 which returns $\hat{\Theta}$. We compare $\hat{\Theta}$ to the ground truth $\Theta$ in $p$ feature space via the distance

$$d(\Theta, \hat{\Theta}) := \|p(\Theta) - p(\hat{\Theta})\|_F. \tag{15}$$

Here, $p$ is the polynomial given in Lemma A.3. Recall from Corollary A.4 that $p$ defines the equivalence class of parameters that compute the same function, i.e., $\mathrm{MHLA}_\Theta = \mathrm{MHLA}_{\hat{\Theta}}$ if and only if $p(\Theta) = p(\hat{\Theta})$. On each dataset, we measure the certificate value $\lambda_{\min}(\Lambda_D)$ on the x-axis vs. $d(\Theta, \hat{\Theta})$ on the y-axis. In Figure 2a, we see that as the certificate value increases, $d(\Theta, \hat{\Theta})$ decreases, indicating that $\mathrm{MHLA}_\Theta$ and $\mathrm{MHLA}_{\hat{\Theta}}$ compute the same function.

**Certifiable Identifiability for MHLA:** Our notion of certifiable identifiability in Lemma A.3 applies to any empirical risk minimizer. Therefore, it applies to popular optimizers like SGD and Adam if they achieve the minimum of the loss, which is in our synthetic case equal to zero. In Figure 2b, we train MHLA models via SGD with $1, 2, 4,$ and $8$ heads. For identifiable data with minimum eigenvalue 0.06, we see that the learned parameters and ground truth parameters are the same in $p$ feature space. However, for unidentifiable data with minimum eigenvalue 0, learned parameters and ground truth parameters are far apart in $p$ feature space and therefore compute different functions.

# 5   Related Work

## 5.1   Formal Expressivity of Transformers

A large body of work has been trying to tackle the problem of quantifying what algorithmic tasks can a Transformer do, in terms of various kinds of circuit families [Pérez et al., 2019, Edelman et al., 2022b, Hahn, 2020, Merrill and Sabharwal, 2023, Merrill et al., 2022, 2021, Liu et al., 2022, Feng et al., 2023]. In particular, researchers have studied how Transformers can realize specific DSLs [Weiss et al., 2021], logic expressions [Dong et al., 2019, Barceló et al., 2020, 2024], Turing machines [Dehghani et al., 2018, Giannou et al., 2023, Pérez et al., 2021], formal language recognition [Hao et al., 2022, Chiang et al., 2023], as well as automata and universal Turing machines [Liu et al., 2022, Li et al., 2024]. However, while these works primarily focus on determining the types of problems whose solutions a Transformer can express, they often overlook the crucial question of how these solutions can be learned from data. Moreover, there is limited discussion on the sufficiency of the dataset itself—whether the data available can identify the underlying "true" function or algorithm that we aim to capture.

## 5.2   Learning Transformers

We break down the literature on learning transformers. First, there is the literature on statistical learnability, where the focus is on the amount of data required to learn without considering whether there is a tractable algorithm for learning [Edelman et al., 2022a, Wei et al., 2021, Zhang et al., 2024, Trauger and Tewari, 2023].

Second, there are learnability results for single head transformers for data distributions under a variety of assumptions. In particular, Zhang et al. [2023] provide learnability results for in-context linear regression; Jelassi et al. [2022] show that data with spatial structure can be learned; the work of Tian et al. [2023] analyzes SGD training dynamics for a toy model for data; and Oymak et al. [2023] study the prompt attention model.

Third, the literature on provable guarantees for learning multi head attention is rather sparse. Fu et al. [2023] give learnability results in a regime where attention matrices are fixed and only the projection matrices are trained. Tarzanagh et al. [2024] show connections between single layer attention optimization and SVM learning. Under a good gradient initialization condition, overparameterization condition, and a condition on the scores of optimal tokens the global convergence of gradient descent to a particular SVM problem can be established. Deora et al. [2023] analyze a setting of learning multi head attention with gradient descent under their Assumption 2. In the words of the authors "these conditions are related to the realizability condition, which guarantees obtaining small training error near initialization", which they instantiate with the separability of the data in an NTK space and a proximity of initialization to realizable parameters. Interestingly, they find that multi head attention has benign optimization properties. Finally, Chen and Li [2024] study learning for multi head attention for well structured data that is drawn independent Bernoulli or Gaussian. They provide an extensive discussion of lower bounds for learning multi head attention.

# 6   Conclusion and Limitations

In this work we tackle the fundamental problem of finding an efficient algorithm that provably learns the weights of a linear Transformer. Our key theoretical ingredient is to consider a model class that's sufficiently "wide" (scaling number of heads), and to find that the loss is convex under this scaling, with generalization guaranteed by the classical VC theory. This reinforces the empirical observation that scaling model size enables efficient optimization and can still result in successful generalization. Our theory extends trivially when arbitrary feature maps $\phi(\cdot)$ are applied to keys and queries providing a natural avenue for extending our theory to models that can approximate softmax transformers with custom key-query kernels. Of course the model class we consider is far simpler than modern LLM's, but we consider our work an important step towards designing algorithms with provable guarantees for training neural sequence models.

**Acknowledgments**

We gratefully acknowledge support from NSF grants IIS-2214177, IIS-2238240, CCF-2112665 and DMS-2134108; from AFOSR grant FA9550-22-1-0249; from ONR MURI grant N00014-22-1-2740; and from ARO grant W911NF-23-1-0034; from the OpenPhilanthropy Foundation; from MIT Quest for Intelligence; from the MIT-IBM Watson AI Lab; from ONR Science of AI; from Simons Center for the Social Brain; and from an Alexander von Humboldt professorship. Any opinions, findings and conclusions or recommendations expressed in this material are those of the authors and do not necessarily reflect the views of our sponsors.

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

# A  Certificate for identifiability of linear attention

We begin by defining identifiability of a model class with respect to a dataset.

**Definition A.1** (Identifiability). Let $D = \{(Z_i, y_i)\}_{i \in [N]}$. Let $\mathcal{U}_\Theta$ denote a model class which is a uniform circuit family parameterized by parameters $\Theta \in \Omega$. Let $\mathcal{L}$ be a loss function and $\Omega_{\text{ERM}}$ be the set of empirical risk minimizers:

$$\Omega_\Theta = \{\hat{\Theta} \in \Omega \mid \hat{\Theta} = \arg\min_{\Theta \in \Omega} \mathcal{L}(\mathcal{U}_\Theta, D)\}. \tag{16}$$

We say model class $\mathcal{U}_\Theta$ is *identifiable with respect to the dataset $D$* if for all $Z \in \mathbb{R}^{d \times n'}$, and for all pairs of empirical risk minimizers $\Theta, \Theta' \in \Omega_{\text{ERM}}$ we have $\mathcal{U}_\Theta$ and $\mathcal{U}_{\Theta'}$ compute the same function, i.e., they agree on all inputs (are the same uniform circuit family):

$$\mathcal{U}_\Theta(Z) = \mathcal{U}_{\Theta'}(Z). \tag{17}$$

In establishing conditions for identifiability, it will be useful to refer to another condition relating models to datasets.

**Definition A.2** (Realizability). Let $\Theta \in \Omega_H$ be an MHLA parameterization. We say a dataset $D = \{(Z_i, y_i)\}_{i \in [N]}$ is *realizable by a parameterization $\Theta$* if $y_i = \text{MHLA}_\Theta(Z_i)$.

The definition of realizability can be modified to include independent noise at the expense of adding some terms to our analyses. See Lemma A.7 for details.

Next, we prove that for the model class MHLA there is an efficiently checkable condition (certificate) of the data $D$ that guarantees the model class is identifiable with respect to $D$. Our results follow by reinterpreting the results of Theorem 2.2 with a focus on data conditions that uniquely determine the optimal regressor. In this section we denote the mapping from data to feature space to be $\mathcal{H}$ and the mapping from parameters to feature space to be $p$ which are analogous to the $X$ and $\mathcal{T}_\Theta$ of Equation (8). We instantiate the feature mapping $\mathcal{H}$ and parameter mapping polynomial $p$ as follows.

**Lemma A.3** (Certificate of Identifiability). *Let dataset $D = \{(Z_i, y_i)\}_{i \in [N]}$ be a realizable dataset. Let $\mathcal{H} = \{\mathcal{H}_n\}_{n=1}^\infty$ be a family of polynomials $\mathcal{H}_n : \mathbb{R}^{d \times n} \to \mathbb{R}^\psi$ for $\psi = \binom{d}{2}d + d^2$ defined as follows. We index the entries of $\mathcal{H}$ by taking the Kronecker product between all sets of pairs $\{j, k\}$ (for all $j, k \in [d]$) with with all $\ell \in [d]$. We define $\mathcal{H}(Z)_{\{j,k\}\ell}$ as in Algorithm 2 to be*

$$\mathcal{H}(Z)_{\{j,k\}\ell} := \langle z_{j:}, z_{k:} \rangle z_{\ell n_i}. \tag{18}$$

*Then if $\lambda_{min}\left(\mathbb{E}_D\left[\mathcal{H}(Z)\mathcal{H}(Z)^T\right]\right) > 0$, we have that $MHLA_\Theta$ is identifiable with respect to $D$.*

Next we construct a mapping $p : \Omega \to \mathbb{R}^{d \times \psi}$ that partitions the parameter space into equivalence classes of parameters that compute the same function. This is akin to matching coefficients in polynomial regression. This mapping defines a meaningful notion of "distance" between different attention parameters by constructing a feature space in which equivalent models have the same representation. We denote the $a$'th row of $p$ to be $p_a : \Omega \to \mathbb{R}^\psi$ and define it as follows.

**Corollary A.4.** *Let $\{p_a\}_{a \in [d]}$ be a collection of polynomials such that $p_a(\Theta) : \Omega_H \to \mathbb{R}^\psi$ is defined as follows. Each $p_a(\Theta)$ is indexed by pairs $\{j, k\}$ for $j, k \in [d]$ and $\ell \in [d]$ defined to be*

$$p_a(\Theta)_{\{j,k\}\ell} = \sum_{h \in [H]} (V_{h,aj}Q_{k\ell} + V_{h,ak}Q_{j\ell}) . \tag{19}$$

*Let the polynomial $p : \Omega \to \mathbb{R}^{d \times \psi}$ be $p := (p_1, p_2, ..., p_d)$. Then for any pair of parameters $\Theta, \Theta' \in \Omega_H$ we have $MHLA_\Theta = MHLA_{\Theta'}$ if and only if $p(\Theta) = p(\Theta')$.*

We give an overview of a few results building on our certifiable identifiability machinery:

First, data drawn from independent noise is certifiably identifiable. If the data matrices $\{Z_i\}_{i \in [N]}$ are drawn with each entry being standard normal noise, then $MHLA_\Theta$ for $\Theta \in \Omega_H$ is identifiable with respect to the data. The statement holds beyond standard normals to distributions satisfying weak moment conditions. The result is stated with population risk instead of empirical risk to simplify the statement.

---

**Algorithm 2** Constructing Features for Certificates of Identifiability

---

1: **Input:** Data $D := \{Z_i\}_{i \in [N]}$ for $Z_i \in \mathbb{R}^{d \times n_i}$
2: **Output:** feature vectors $\mathcal{H}(Z_i)$ for $i \in [N]$
3: **for** $Z_i \in D$ **do**
4:      Let $z_{1:}, z_{2:}, \ldots z_{d:}$ be the rows of $Z_i$ and let $z_{ab}$ be the $(a,b)$ entry of $Z_i$
5:      **for** sets $\{j, k\}$ in Distinct Pairs of Indices in $[d]^2$ **do**
6:         **for** $\ell \in [d]$ **do**
7:            $\mathcal{H}(Z_i) = \mathcal{H}(Z_i) \circ [\langle z_{j:}, z_{k:} \rangle z_{\ell n_i}]$
8:         **end for**
9:      **end for**
10:     **for** $j \in [d]$ **do**
11:        **for** $\ell \in [d]$ **do**
12:           $\mathcal{H}(Z_i) = \mathcal{H}(Z_i) \circ \left[\|z_j\|^2 z_{\ell n_i}\right]$
13:        **end for**
14:     **end for**
15: **end for**
16: **Return:** $\{\mathcal{H}(Z_i)\}_{i \in [N]}$

---

**Lemma A.5** (Independent input noise yields identifiability). *Let $(Z, y) \sim \mathcal{D}$ be a realizable dataset. Let $Z$ be drawn from a distribution $\mathcal{Z}$ where the $(a, b)$-th entry of $Z$ denoted by $Z_{ab}$ is drawn i.i.d. from a distribution $\nu$ over $\mathbb{R}$ for all $a \in [d]$ and $b \in [n]$. Let the second and fourth moment of $\nu$ be denoted $m_2$ and $m_4$ respectively. Let $m_2 > 0$ and $m_4 > m_2^2$. Then $MHLA_\Theta$ for $\Theta \in \Omega_H$ is identifiable with respect to $D$. That is to say, for any population risk minimizers $\Theta, \Theta' \in \Omega_{PRM}$:*

$$MHLA_\Theta = MHLA_{\Theta'}. \tag{20}$$

Second, when specialized to the case of Multi Head Linear Attention $MHLA_\Theta$ with more than $d^2$ heads we can avoid the realizability assumption entirely. This is because the class of MHLA with an arbitrary number of heads is linear in the feature space $\mathcal{H}$ given in Lemma A.3.

**Lemma A.6** (Identifiability without realizability for MHLA with arbitrarily many heads). *Let dataset $D = \{(Z_i, y_i)\}_{i \in [N]}$ be any dataset drawn i.i.d from a distribution $\mathcal{D}$. Let $\mathcal{H}$ be defined as in Lemma A.3. Then if $\lambda_{\min}\left(\mathbb{E}_D[\mathcal{H}(Z)\mathcal{H}(Z)^T]\right) > 0$ then $MHLA_\Theta$ for $\Theta \in \Omega_H$ for any $H \in [d^2, \infty)$ is identifiable with respect to the data $D$. That is,*

$$MHLA_\Theta = MHLA_{\Theta'} \tag{21}$$

*for all pairs of empirical risk minimizers $\Theta, \Theta' \in \Omega_{ERM}$.*

We also add a quantitative version of identifiability with precise treatment of issues related to error. (For a corresponding statement of realizability with noise see Lemma E.2.)

**Lemma A.7** (Identifiability with Error). *Let $\Omega_{\epsilon-ERM}$ be the set of $\epsilon$-approximate empirical risk minimizers,*

$$\Omega_{\epsilon-ERM} =$$
$$\left\{ \Theta \in \Omega_H \mid \mathbb{E}_{(Z_i, y_i) \in D}\left[ (MHLA_\Theta(Z_i) - y_i)^2 \right] \leq \epsilon \right\}.$$

*Then we have for any $\Theta, \Theta' \in \Omega_{\epsilon-ERM}$ that for all inputs $Z \in \mathbb{R}^{d \times n}$*

$$\|MHLA_\Theta(Z) - MHLA_{\Theta'}(Z)\| \leq \frac{\epsilon}{\lambda_{\min}(\Lambda_D)} \|Z\|_F^6. \tag{22}$$

We prove all the above statements in Appendix E.

**Application to learning Universal Turing Machines.** In Appendix B, we demonstrate that MHLAs can (autoregressively) express universal Turing machines with polynomially bounded computation histories. In this context, our identifiability results imply that, given a certifiably identifiable dataset of Turing machines and their computation histories on input words, empirical risk minimization and in particular Algorithm 1 will learn the universal Turing machine in a strong sense (Lemma C.5 for learning, Lemma A.8 with identifiability). That means at test time the learned MHLA will simulate any Turing Machine on any input word up to a given size for a bounded number of steps. For more detail see C

**Lemma A.8** (Learning UTM from Certifiably Identifiable Data). *Let $D = \{(Z_i, y_i)\}_{i \in [N]}$ be a dataset satisfying $y_i = MHLA_\Theta$ for $\Theta \in \Omega_H$ being the expressibility parameters of Lemma B.1 for the set of TM's/words $(M, x) \in \Delta(\hat{\mathcal{Q}}, \hat{\Sigma}, \hat{n}, \hat{\Phi})$. If $D$ is certifiably identifiable with $\lambda_{min}(\Lambda_D) > \eta$, then there is a $poly(d, N, \hat{Q}, \hat{\Sigma}, \hat{n}, \hat{\Phi}, \eta^{-1})$ time algorithm that outputs a set of parameters $\hat{\Theta} \in \Omega_{d^2}$ such that for all TM's $M$ and input words $x$ in $\Delta(\hat{\mathcal{Q}}, \hat{\Sigma}, \hat{n}, \hat{\Phi})$, we have*

$$CH_{\hat{\Theta}}(M, x)^{c(t)}[: -k_t] = x^t . \tag{23}$$

*The $c(t)$ step of the autoregressive computation history of $\hat{\Theta}$ is equal to the $t$'th step of the computation history of $M$ on $x$.*

## B  Realizability of Universal Automata in MHLA

We also include an application of our theory on learnability and identifiability to the problem of learning a universal Turing machine (UTMs) with polynomially bounded computation length. We prove such a UTM is expressible via MHLA in Lemma B.1, and show that for certifiably identifiable data the learned MHLA generalizes to any TM $M$ and input word $x$ in Lemma A.8.

**Lemma B.1** (UTM Expressibility). *Let $\Delta(\hat{\mathcal{Q}}, \hat{\Sigma}, \hat{n}, \hat{\Phi})$ be the set of Turing machines $M = \{\delta, \Sigma, \mathcal{Q}, q_{start}, q_{accept}, q_{reject}\}$ and words $x \in \Sigma^*$ with number of states, size of alphabet, size of input, and number of steps in computation history bounded by $\hat{\mathcal{Q}}, \hat{\Sigma}, \hat{n}, \hat{\Phi}$ respectively. For any $(M, x) \in \Delta$, let $\{x_t\}_{t \in [\Phi]}$ be the computation history of the UTM on $(M, x)$. Let the autoregressive computation history (see Definition C.2) of $MHLA_\Theta$ on input $(M, x)$ be denoted $CH_\Theta(M, x) = \{Z^1, Z^2, ..., Z^\Phi\}$. Then there exists a set of parameters $\Theta \in \Omega_H$ for $H = O(\hat{n}\hat{\Phi}\hat{\Sigma})$ and embedding dimension $d = O(\hat{n}\hat{\Phi}\hat{\Sigma} \max(\hat{\Sigma}, \hat{\mathcal{Q}}))$, such that for all $(M, x) \in \Delta$, the TM computation history at time step $t$ is equivalent to the autoregressive computation history at time step $c(t)$ where $c(t) \leq O((n + t)t)$ i.e $Z^{c(t)}[: -length(x^t)] = x^t$. Furthermore, this can be achieved with 2 bits of precision.*

Our construction bears similarities to [Pérez et al., 2019, Hahn, 2020, Merrill and Sabharwal, 2023, Merrill et al., 2022, 2021, Liu et al., 2022, Feng et al., 2023]; the high-level idea is write down every letter in the computation history of $M$ on $x$. If we use orthogonal vectors to encode every letter, state, and positional embedding we arrive at a natural construction involving a few basic primitives copy, lookup, and if-then-else. For details see discussion section F and Proof F.1

## C  Application to Learning Universal Turing Machines

We apply our algorithmic and identifiability machinery to show that an important computational procedure is representable and learnable as an MHLA: namely, a restricted class of universal Turing machines (UTMs) with bounded computation history. We must first generalize our previous MHLA definition to enable multi-step computation:

**Definition C.1** (Autoregressive MHLA). Let $Z^0$ be an input matrix in dimension $\mathbb{R}^{d \times n}$. We define the iterative process of $\Phi$-*step autoregressive MHLA* as follows: starting from $t = 0$, let the next token $y^{t+1} \in \mathbb{R}^d$ be:

$$y^{t+1} = MHLA_\Theta(Z^t) , \tag{24}$$

and, for all $t \in [\Phi]$, let $Z^{t+1} \in \mathbb{R}^{d \times (n+1)}$ be the concatenation:

$$Z^{t+1} = Z^t \circ y^t . \tag{25}$$

Next we define the computation history of an autoregressive model analogously to the computation history of a Turing machine.

**Definition C.2** (Autoregressive Computation History). We refer to $CH_\Theta(Z) = \{Z^t\}_{t \in [\Phi]}$ as the *computation history* of the $\Phi$-step autoregressive MHLA. We denote the $t$-th step of the computation history as $CH_\Theta^t(Z) = Z^t$.

We will often use the notation $Z_t[: -k]$ to denote the last $k \in \mathbb{Z}^+$ tokens of $Z_t$. Often, $Z$ will be the embeddings corresponding to a word $x$ in a language $\mathcal{L}$, in which case we will use the notation

$CH_\Theta(x)$ and $CH_\Theta(Z)$ interchangeably. For pedagogical discussion on how to map embeddings to letters in an alphabet, see Section G

Although the theory derived in this paper applies to all functions expressible by MHLAs, we are particularly interested in the task of learning *universal Turing machines* (UTMs). Let $\Sigma$ be an alphabet. Let $\mathcal{Q}$ be a set of states that includes $\{q_{start}, q_{accept}, q_{reject}\}$ a start, accept, and reject state respectively. Let $\delta : \mathcal{Q} \times \Sigma \to \mathcal{Q} \times \Sigma \times \{L/R\}$ be a transition function that takes an alphabet and state symbol and maps to a state transition, an output symbol, and a head movement left or right. Typically there is also a tape alphabet $\Gamma$ for which the input alphabet $\Sigma$ is a subset.

**Definition C.3** (Accept TM). Let $M = \{\delta, \Sigma, \Gamma, \mathcal{Q}, q_{start}, q_{accept}, q_{reject}\}$ be a TM. Let $x \in \Sigma^*$ be all strings in the alphabet $\Sigma$. Then let $A_{\text{TM}}$ be the language $A_{\text{TM}} = \{(M, x) \mid M \text{ accepts } x\}$.

The UTM constructed in Turing's 1936 paper recognizes $A_{\text{TM}}$. In practice, we are most often interested in the behavior of TMs that run in polynomial time, and focus below on implementing a universal simulator for this restricted class:

**Definition C.4.** (Polynomially Bounded Universal Turing Machine) In general, a UTM is a recognizer for the language $A_{\text{TM}}$. That is if $x$ is in $A_{\text{TM}}$, the UTM accepts, else, the UTM rejects or does not halt. Let $A_{\text{TM}} \cap P$ be the language of input pairs $(M, x)$ for TM $M$ and word $x \in \Sigma^*$ such that $M$ decides $x$ in polynomial time. Here, we consider UTM to be the polynomial time decider for $A_{\text{TM}} \cap P$.

To define what it means for an autoregressive MHLA to perform the same computation as a TM, our main idea is to construct parameters for MHLA such that it executes the computation history of TM $M$ on input $x$. Let the UTM computation history at step $t$ include the contents $x_0, \dots, x_{k_t}$ on the tape after $t$ transition steps of the Turing machine $M$, the current state $q_t$, and the current head position $h_t$. Here $k_t$ is the number of tokens at timestep $t$. Then, there is a single-layer MHLA capable of simulating a UTM:

**Lemma B.1** (UTM Expressibility). *Let $\Delta(\hat{\mathcal{Q}}, \hat{\Sigma}, \hat{n}, \hat{\Phi})$ be the set of Turing machines $M = \{\delta, \Sigma, \mathcal{Q}, q_{start}, q_{accept}, q_{reject}\}$ and words $x \in \Sigma^*$ with number of states, size of alphabet, size of input, and number of steps in computation history bounded by $\hat{\mathcal{Q}}, \hat{\Sigma}, \hat{n}, \hat{\Phi}$ respectively. For any $(M, x) \in \Delta$, let $\{x_t\}_{t \in [\Phi]}$ be the computation history of the UTM on $(M, x)$. Let the autoregressive computation history (see Definition C.2) of MHLA$_\Theta$ on input $(M, x)$ be denoted $CH_\Theta(M, x) = \{Z^1, Z^2, ..., Z^\Phi\}$. Then there exists a set of parameters $\Theta \in \Omega_H$ for $H = O(\hat{n}\hat{\Phi}\hat{\Sigma})$ and embedding dimension $d = O(\hat{n}\hat{\Phi}\hat{\Sigma} \max(\hat{\Sigma}, \hat{\mathcal{Q}}))$, such that for all $(M, x) \in \Delta$, the TM computation history at time step $t$ is equivalent to the autoregressive computation history at time step $c(t)$ where $c(t) \leq O((n + t)t)$ i.e $Z^{c(t)}[: -length(x^t)] = x^t$. Furthermore, this can be achieved with 2 bits of precision.*

We include the full proof for the existence of $\Theta$ in the appendix. For simplicity, we adopt a naive embedding scheme that represents different letters in an alphabet as orthogonal unit vectors. This makes it easy to contrive embedding schemes that incorporate arbitrary polynomial-sized circuits which could compute whether $x \in \mathcal{L}(M)$. Moreover, we adopt positional encodings that are simply orthogonal unit vectors. Thus, in order to give each of $T$ tokens a unique ID, we would require $O(T)$ dimensional positional embeddings.

This can be combined with the learnability results above to yield a specialized result for UTMs:

**Lemma C.5** (Learning a UTM). *Let $\Theta \in \Omega_H$ in dimension $d$ be the MHLA parameters in Lemma B.1. Let $\{M_i, x_i\}_{i \in [N]}$ be pairs of TM's $M$ and words $x$ of maximum length $n$ drawn i.i.d. from a distribution $\mathcal{D}$. Let $Z_i = Embed(M_i, x_i)$. For each TM/word pair $(M_i, x_i)$ let $CH_\Theta(Z_i) = \{Z_i^1, Z_i^2, ..., Z_i^\Phi\}$ be the $\Phi$-step autoregressive computation history of MHLA$_\Theta$ on $Z_i$. Let $D$ be the dataset $D := \{(CH_\Theta(Z_i)^t, y_i^{t+1}\}_{i \in [N], t \in [T]}$ where $y_i^{t+1} = MHLA_\Theta(Z_i^t)$. Then Algorithm 1 applied to input $D$ returns $\hat{\Theta} \in \Omega_H$ for $H \leq d^2$ such that with probability $1 - \delta$*

$$\mathbb{E}_{(Z,y) \in \mathcal{D}} \left[ \left( MHLA_{\hat{\Theta}}(Z) - y \right)^2 \right] \leq \epsilon \tag{26}$$

*for sample complexity $N = poly(d, \epsilon^{-1}, \log(\delta^{-1}))$. Then with probability $1 - \delta$ over the randomness in the data, the probability over $\mathcal{D}$ that the $\Phi$-step autoregressive computation history $CH_{\hat{\Theta}}(M, x)$ and $CH_\Theta(M, x)$ differ is upper bounded by*

$$\Pr_{(M,x) \sim \mathcal{D}}[CH_{\hat{\Theta}}(M, x) \neq CH_\Theta(M, x)] \leq O(\epsilon\Phi). \tag{27}$$

Finally, if the dataset $D$ is certifiably identifiable, then generalization holds out-of-distribution. For proof see Appendix F.2.

**Lemma A.8** (Learning UTM from Certifiably Identifiable Data). *Let $D = \{(Z_i, y_i)\}_{i \in [N]}$ be a dataset satisfying $y_i = MHLA_\Theta$ for $\Theta \in \Omega_H$ being the expressibility parameters of Lemma B.1 for the set of TM's/words $(M, x) \in \Delta(\hat{\mathcal{Q}}, \hat{\Sigma}, \hat{n}, \hat{\Phi})$. If $D$ is certifiably identifiable with $\lambda_{min}(\Lambda_D) > \eta$, then there is a $poly(d, N, \hat{\mathcal{Q}}, \hat{\Sigma}, \hat{n}, \hat{\Phi}, \eta^{-1})$ time algorithm that outputs a set of parameters $\hat{\Theta} \in \Omega_{d^2}$ such that for all TM's $M$ and input words $x$ in $\Delta(\hat{\mathcal{Q}}, \hat{\Sigma}, \hat{n}, \hat{\Phi})$, we have*

$$CH_{\hat{\Theta}}(M, x)^{c(t)}[: -k_t] = x^t . \tag{23}$$

*The $c(t)$ step of the autoregressive computation history of $\hat{\Theta}$ is equal to the $t$'th step of the computation history of $M$ on $x$.*

# D  Proof of the Main Theorem

**Theorem 2.2** (Learnability of Linear Attention). *Let $D$ be a dataset $D = \{Z_i, y_i\}_{i \in [N]}$ drawn i.i.d. from a distribution $\mathcal{D}$ where each $Z_i \in \mathbb{R}^{d \times n_i}$, $y_i \in \mathbb{R}^d$. The embedding dimension $d$ is fixed across the dataset, whereas $n_i$ can be different for each datapoint. Let $n_{max} = \max_{i \in [N]} n_i$ be the maximum sequence length, and let $\Omega_H$ be the space of $H$ pairs of value and key-query matrices $\{(V_h, Q_h)\}_{h \in [H]}$ for any $H \in [1, \infty)$. Then there is an algorithm (Algorithm 1) that runs in time $O(Nd^4 n_{max} \epsilon^{-1})$ and that, given input–output pairs $\{(Z_i, y_i)\}_{i \in [N]}$, returns $\hat{\Theta} = \{(\hat{V}_h, \hat{Q}_h)\}_{h \in [\hat{H}]} \in \Omega_{\hat{H}}$ for $\hat{H} \le d^2$ such that with probability $1 - \delta$,*

$$\mathbb{E}_{(Z,y) \in \mathcal{D}} \left[ \|MHLA_{\hat{\Theta}}(Z) - y\|^2 \right]$$
$$- \min_{\Theta \in \Omega_H} \mathbb{E}_{(Z,y) \in \mathcal{D}} \left[ \|MHLA_\Theta(Z) - y\|^2 \right] \le \epsilon \tag{6}$$

*with sample complexity $N = O\left( \frac{1}{\epsilon} \left( d^4 + \log(\delta^{-1}) \right) \right)$.*

*Proof.* First we write down the loss:

$$\mathcal{L}_\Theta(\{(Z_i, y_i)\}_{i \in [N]}) := \frac{1}{N} \sum_{i \in [N]} \left\| \sum_{h \in [H]} V_h Z_i (Z_i^T Q_h Z[:, n_i]) - y_i \right\|_F^2 \tag{28}$$

$$= \frac{1}{N} \sum_{i \in [N]} \sum_{a \in [d]} \left( \sum_{h \in [H]} e_a^T V_h Z_i (Z_i^T Q_h Z[:, n_i]) - y_{i,a} \right)^2 \tag{29}$$

Observe that the one layer attention network is a quadratic polynomial in $\{V_h, Q_h\}_{h \in [H]}$.

$$= \frac{1}{N} \sum_{i \in [N]} \sum_{a \in [d]} (\langle \mathcal{T}_\Theta, X_{i,a} \rangle - y_{i,a})^2 \tag{30}$$

Here

$$\mathcal{T}_\Theta := \sum_{h \in [H]} \text{flatten}(V_h) \text{flatten}(Q_h)^T = \sum_{h \in [H]} \begin{bmatrix} V_{h,00}Q_{h,00} & V_{h,00}Q_{h,01} & \cdots & V_{h,00}Q_{h,dd} \\ V_{h,01}Q_{h,00} & V_{h,01}Q_{h,01} & \cdots & V_{h,01}Q_{h,dd} \\ \vdots & \vdots & \vdots \\ V_{h,dd}Q_{h,00} & V_{h,dd}Q_{h,01} & \cdots & V_{h,dd}Q_{h,dd} \end{bmatrix} \tag{31}$$

Now we relax the objective where we replace $\mathcal{T}_\Theta$ with an unconstrained matrix $W \in \mathbb{R}^{d^2 \times d^2}$. Another way to put it is that $\mathcal{T}_\Theta$ is rank-$H$ but $W$ can be a general matrix. Because the space of general rank matrices is larger, we have written down a relaxation guaranteed to have a smaller loss. Furthermore the loss can be optimized via ordinary least squares.

$$\min_{W \in \mathbb{R}^{d^2 \times d^2}} \mathcal{L}_W(\{(Z_i, y_i)\}_{i \in [N]}) := \frac{1}{N} \sum_{i \in [N]} \sum_{a \in [d]} (\langle W, X_{i,a} \rangle - y_{i,a})^2$$

$$\leq \min_{\Theta \in \Omega_H} \mathcal{L}_\Theta(\{(Z_i, y_i)\}_{i \in [N]}) + \epsilon \quad (32)$$

Thus the optimum of the regression with respect to the data achieves optimum of the loss to error $\epsilon$ in time $O(\frac{1}{\epsilon} d^4 N)$. The sample complexity to achieve error $\epsilon$ is then $O(\frac{1}{\epsilon}(d^4 + \log(\delta^{-1})))$ with probability $1 - \delta$ over the data distribution. Furthermore, if we take the SVD of $W = \sum_{i \in [\hat{H}]} A_i B_i^T$ where we absorb the singular values into the left and right singular vectors we have for $\hat{\Theta} = \{\text{Fold}(A_h), \text{Fold}(B_h)\}_{i \in [\hat{H}]}$. Let $\hat{V}_h = \text{Fold}(A_h)$ and $\hat{Q}_h = \text{Fold}(B_h)$

$$\mathcal{L}_{\hat{\Theta}}(\{(Z_i, y_i)\}_{i \in [N]}) := \frac{1}{N} \sum_{i \in [N]} \left\| \sum_{h \in [\hat{H}]} \hat{V}_h Z_i (Z_i^T \hat{Q}_h Z_i[:, n_i]) - y_i \right\|_F^2$$

$$= \frac{1}{N} \sum_{i \in [N]} \sum_{a \in [d]} \left( \sum_{h \in [\hat{H}]} \hat{V}_h Z_i (Z_i^T \hat{Q}_h Z_i[:, n_i]) - y_{i,a} \right)^2 \leq \epsilon \quad (33)$$

as desired. $\qquad \square$

## E   Proofs from Identifiability Section

First, we start with a general lemma (Lemma E.1) which states a sufficient condition for identifiability of any model class that can be written as an inner product of a polynomial of parameters $\Theta$ with a polynomial feature mapping $\mathcal{H}$. If the data is realizable by the model class and $\Lambda_D = \mathbb{E}_D \left[ \mathcal{H}(Z)\mathcal{H}(Z)^T \right]$ is full rank then the model class is identifiable with respect to $D$.

The following is the certificate of identifiability written in an abstract form involving polynomials to map parameters to feature space and polynomials to map data to feature space. The proof does not require the model to be an MHLA, but we state it in MHLA terms for the sake of concreteness.

**Lemma E.1** (General Certificate of Identifiability). *Let dataset $D = \{(Z_i, y_i)\}_{i \in [N]}$ be a dataset realizable by $\Theta \in \Omega_H$. Let $p := \{p_a\}_{a \in [d]}$ be a collection of polynomials $p_a : \Omega \to \mathbb{R}^\psi$ mapping the parameters $\Theta \in \Omega$ to a feature space of fixed dimension $\psi \in \mathbb{Z}^+$. Let $\mathcal{H} = \{\mathcal{H}_n\}_{n=1}^\infty$ be a uniform family of polynomials such that $\mathcal{H}_n : \mathbb{R}^{d \times n} \to \mathbb{R}^\psi$. Let $p$ and $\mathcal{H}$ satisfy*

$$MHLA_\Theta(Z)[a] = \langle p_a(\Theta), \mathcal{H}_n(Z) \rangle \quad (34)$$

*for all $Z \in \mathbb{R}^{d \times n}$ for all $n \in [1, \infty)$. Then if $\lambda_{\min} \left( \mathbb{E}_D \left[ \mathcal{H}(Z)\mathcal{H}(Z)^T \right] \right) > 0$, we have*

$$MHLA_\Theta = MHLA_{\Theta'} \quad (35)$$

*for all empirical risk minimizers $\Theta, \Theta' \in \Omega_{ERM}$. That is, all empirical risk minimizers compute the same function.*

*Proof.* We construct a map $p : \Omega \to \mathbb{R}^\psi$ such that $MHLA_\Theta = MHLA_{\Theta'}$ if and only if $p(\Theta) = p(\Theta')$. Then we show that any empirical risk minimizer $\Theta_{ERM}$ and the ground truth $\bar{\Theta}$ satisfy $p(\Theta_{ERM}) = p(\bar{\Theta})$.

In more detail, we construct some polynomials $\{p_a\}_{a \in [d]}$ and family of polynomials $\mathcal{H}$ such that

$$\text{MHLA}_\Theta(Z)|_a = \langle p_a(\Theta), \mathcal{H}(Z) \rangle \quad (36)$$

We construct a linear model class $\mathcal{R}$ that takes as parameters $v \in \mathbb{R}^\psi$ and data $\mathcal{H}(Z) \in \mathbb{R}^\psi$. such that

$$\mathcal{R}_v(\mathcal{H}(Z)) = \langle v, \mathcal{H}(Z) \rangle \quad (37)$$

Let $\Theta_{\text{ERM}}$ be defined as

$$\Theta_{\text{ERM}} := \{\Theta' \in \Omega | \Theta' = \underset{\Theta \in \Omega}{\arg\min} \, \mathbb{E}_{i \in [N]} \left[ \mathcal{L}(\text{MHLA}_\Theta(Z_i), y_i) \right] \} \tag{38}$$

Let $v_{\text{ERM}}$ be defined as

$$v_{\text{ERM}} := \{v' \in \mathbb{R}^\psi | v' = \underset{v \in \mathbb{R}^\psi}{\arg\min} \, \mathbb{E}_{i \in [N]} \left[ \mathcal{L}(\mathcal{R}_v(\mathcal{H}(Z_i)), y_i) \right] \} \tag{39}$$

Observe that for all $\Theta \in \Theta_{\text{ERM}}$, we have $p(\Theta) \subseteq v_{\text{ERM}}$. Here we use the fact that $y$ is realizable by the ground truth $\bar{\Theta}$. Therefore if we show that $v_{\text{ERM}}$ is unique, i.e comprised of a single element then $p_{\text{ERM}} := \{p(\Theta) | \Theta \in \Theta_{\text{ERM}}\}$ is also unique. Therefore, $\text{MHLA}_\Theta$ is the same function for any $\Theta \in \Theta_{\text{ERM}}$

To show $v_{\text{ERM}}$ is unique, all we need is that the second moment of the features $\Lambda_D = \mathbb{E}_D \left[ \mathcal{H}(Z)\mathcal{H}(Z)^T \right]$ is positive definite (the covariance has a minimum eigenvalue bounded away from zero). $\qquad\square$

Next we prove the main certifiable identifiability lemma by instantiating the polynomials $\mathcal{H}$ and $p$ from Lemma E.1.

**Lemma A.3** (Certificate of Identifiability). *Let dataset $D = \{(Z_i, y_i)\}_{i \in [N]}$ be a realizable dataset. Let $\mathcal{H} = \{\mathcal{H}_n\}_{n=1}^\infty$ be a family of polynomials $\mathcal{H}_n : \mathbb{R}^{d \times n} \to \mathbb{R}^\psi$ for $\psi = \binom{d}{2}d + d^2$ defined as follows. We index the entries of $\mathcal{H}$ by taking the Kronecker product between all sets of pairs $\{j, k\}$ (for all $j, k \in [d]$) with with all $\ell \in [d]$. We define $\mathcal{H}(Z)_{\{j,k\}\ell}$ as in Algorithm 2 to be*

$$\mathcal{H}(Z)_{\{j,k\}\ell} := \langle z_{j:}, z_{k:} \rangle z_{\ell n_i}. \tag{18}$$

*Then if $\lambda_{min} \left( \mathbb{E}_D \left[ \mathcal{H}(Z)\mathcal{H}(Z)^T \right] \right) > 0$, we have that $\text{MHLA}_\Theta$ is identifiable with respect to $D$.*

*Proof.* First we construct a polynomial $p : \Omega \to \mathbb{R}^\psi$ and $\mathcal{H} : \mathbb{R}^{d \times n} \to \mathbb{R}^\psi$ for $\psi = \binom{d}{2}d + d^2$ such that

$$\text{MHLA}_\Theta(Z)[a] = \langle p_a(\Theta), \mathcal{H}(Z) \rangle \tag{40}$$

We begin by rewriting $\text{MHLA}_\Theta(Z)[a]$. We index the first $\binom{d}{2}d$ entries of $p_a(\Theta)$ by all pairs $\{j, k\}$ for $j, k \in [d]$ and all $\ell \in [d]$.

$$p_a(\Theta)_{\{j,k\},\{\ell\}} := \sum_{h \in [H]} \left( V_{h,aj}Q_{h,k\ell} + V_{h,ak}Q_{h,j\ell} \right) \tag{41}$$

We define the entries of $p_a(\Theta)$ from $[\binom{d}{2}d, \binom{d}{2}d + d^2]$ as follows.

$$p_a(\Theta)_{\{j^2\}\{\ell\}} := \sum_{h \in [H]} V_{h,aj}Q_{h,j\ell} \tag{42}$$

Similarly, we define $\mathcal{H}(Z)$ be be the following $\binom{d}{2}d + d^2$ features. $\mathcal{H}(Z)_{\{j,k\}\{\ell\}}$ and $\mathcal{H}(Z)_{\{\ell\}}$.

$$\mathcal{H}(Z)_{\{j,k\}\{\ell\}} := \langle z_{j:}, z_{k:} \rangle z_{\ell n} \tag{43}$$

and

$$\mathcal{H}(Z)_{\{j^2\}\{\ell\}} := \|z_{j:}\|^2 z_{\ell n} \tag{44}$$

Thus we rewrite $\text{MHLA}_\Theta(Z)[a]$ as

$$\text{MHLA}_\Theta(Z)[a] = \sum_{\{j,k\} \in \mathcal{S}_2^d} \sum_{\ell \in [d]} p_a(\Theta)_{\{j,k\},\{\ell\}}\mathcal{H}(Z)_{\{j,k\}\{\ell\}} + \sum_{j,\ell \in [d]} p_a(\Theta)_{\{j^2\}\{\ell\}}\mathcal{H}(Z)_{\{j^2\}\{\ell\}}$$

$$= \langle p_a(\Theta), \mathcal{H}(Z) \rangle \tag{45}$$

Here we introduce the notation $\mathcal{S}_2^d$ to denote the set of all pairs $\{j, k\}$ for $j, k \in [d]$. We have constructed a polynomial $p_a(\Theta)$ such that for any $\Theta, \Theta' \in \Omega$ in the same equivalence class $p_a(\Theta) = p_a(\Theta')$, we have $\text{MHLA}_\Theta = \text{MHLA}_{\Theta'}$. Furthermore, if there exists $b \in [n]$ such that $\lambda_{min} \left( \mathbb{E}_D \left[ \mathcal{H}(Z)\mathcal{H}(Z)^T \right] \right) > 0$ then OLS returns a unique solution for $p_a(\Theta)$. Since the data is realizable, we conclude $p_a(\Theta) = p_a(\bar{\Theta})$ for all $\Theta \in \Omega_{\text{ERM}}$. $\qquad\square$

Next we present the proof that realizability is not necessary to identify the function learned by MHLA with more than $d^2$ heads.

**Lemma A.6** (Identifiability without realizability for MHLA with arbitrarily many heads)**.** *Let dataset $D = \{(Z_i, y_i)\}_{i \in [N]}$ be any dataset drawn i.i.d from a distribution $\mathcal{D}$. Let $\mathcal{H}$ be defined as in Lemma A.3. Then if $\lambda_{\min}\left(\mathbb{E}_D[\mathcal{H}(Z)\mathcal{H}(Z)^T]\right) > 0$ then $MHLA_\Theta$ for $\Theta \in \Omega_H$ for any $H \in [d^2, \infty)$ is identifiable with respect to the data $D$. That is,*

$$MHLA_\Theta = MHLA_{\Theta'} \tag{21}$$

*for all pairs of empirical risk minimizers $\Theta, \Theta' \in \Omega_{ERM}$.*

*Proof.* We know from [lemma main algorithm] there exists a surjective map $p_a(\Theta)$ that takes $\Theta \in \Omega$ into $v \in \mathbb{R}^\psi$. This implies that for all $v \in \mathbb{R}^\psi$ there exists a right inverse function $p^r(v) = \Theta$ satisfying $p(\Theta) = v$ given by SVD. Therefore, $p(\Theta_{ERM}) \in v_{ERM}$ i.e optimizing over $v \in \mathbb{R}^\psi$ does no better than optimizing over $\Theta \in \Omega$. To prove this consider the contrary that there exists $v' \in v_{ERM}$ and there is no $\Theta \in \Omega$ that achieves the same empirical risk as $v'$. However, $p^r(v) \in \Omega$ is such a $\Theta$, and we have a contradiction. The key point is that we avoid the assumption of realizability and replace it with surjectivity of the polynomials $p_a$. $\qquad\square$

Finally we prove that data drawn from independent noise is certifiably identifiable. A subtlety in the proof is that we use a somewhat different set of polynomials than Lemma A.3 as we center and normalize our features, which still satisfies the assumptions of the general certificate Lemma E.1

**Lemma A.5** (Independent input noise yields identifiability)**.** *Let $(Z, y) \sim \mathcal{D}$ be a realizable dataset. Let $Z$ be drawn from a distribution $\mathcal{Z}$ where the $(a, b)$-th entry of $Z$ denoted by $Z_{ab}$ is drawn i.i.d. from a distribution $\nu$ over $\mathbb{R}$ for all $a \in [d]$ and $b \in [n]$. Let the second and fourth moment of $\nu$ be denoted $m_2$ and $m_4$ respectively. Let $m_2 > 0$ and $m_4 > m_2^2$. Then $MHLA_\Theta$ for $\Theta \in \Omega_H$ is identifiable with respect to $D$. That is to say, for any population risk minimizers $\Theta, \Theta' \in \Omega_{PRM}$:*

$$MHLA_\Theta = MHLA_{\Theta'}. \tag{20}$$

*Proof.* We give the entries of $\Lambda(Z)$ the following naming convention. Let the terms $\{j, k\}\{\ell\}$ and pairs $\{j', k'\}\{\ell'\}$. Terms that involve $\{j^2\}\{\ell\}$ and $\{j'^2\}\{\ell'\}$ are referred to as 'singles'.

$$\mathbb{E}\left[\mathcal{H}_b(Z)_{\{j,k\}\{\ell\}}\mathcal{H}_b(Z)_{\{j',k'\}\{\ell'\}}\right] = \frac{1}{n}\mathbb{E}[\langle z_{j:}, z_{k:}\rangle\langle z_{j':}, z_{k':}\rangle z_{\ell b} z_{\ell' b}] \tag{46}$$

We give entries of the following form the name "singles to singles"

$$\mathbb{E}\left[\mathcal{H}_b(Z)_{\{j^2\}\{\ell\}}\mathcal{H}_b(Z)_{\{j'^2\}\{\ell'\}}\right] = \frac{1}{n}\mathbb{E}[(\|z_{j:}\|^2 - nm_2)(\|z_{j':}\|^2 - nm_2)z_{\ell b}^2] \tag{47}$$

For the case of $Z$ drawn with each entry i.i.d $\nu$ we can proceed via case work.

**Case 1: Pairs to Pairs, $j \neq k$ and $j' \neq k'$**

1. **Subcase 1: $\{j, k\} \neq \{j', k'\}$ and $\ell = \ell'$:**
$$\frac{1}{n}\mathbb{E}[\langle z_{j:}, z_{k:}\rangle\langle z_{j':}, z_{k':}\rangle z_{\ell b} z_{\ell' b}] = 0 \tag{48}$$

2. **Subcase 2: $\{j, k\} = \{j', k'\}$ and $\ell = \ell'$:**
$$\frac{1}{n}\mathbb{E}[\langle z_{j:}, z_{k:}\rangle^2 z_{\ell b}^2] = m_2^3 \tag{49}$$

**Case 2: Singles to Singles, $j = k$ and $j' = k'$**

1. **Subcase 1: $j \neq j'$ and $\ell = \ell'$:**
$$\frac{1}{n}\mathbb{E}\left[\left(\|z_{j:}\|^2 - nm_2\right)\left(\|z_{j':}\|^2 - nm_2\right)z_{\ell b}^2\right] = 0 \tag{50}$$

2. **Subcase 2: $j = j'$ and $\ell = \ell'$:**
$$\frac{1}{n}\mathbb{E}\left[\left(\|z_{j:}\|^2 - nm_2\right)^2 z_{\ell b}^2\right] = \frac{1}{n}\left((n^2 - n)m_2^2 + nm_4 - n^2 m_2^2\right)m_2 = (m_4 - m_2^2)m_2 \tag{51}$$

**Case 3: Singles to Pairs,** $j = k$ **and** $j' \neq k'$

1. **Subcase 1:** $\ell = \ell'$:

$$\frac{1}{n} \mathbb{E}\left[\left(\|z_{j:}\|^2 - nm_2\right) \langle z_{j':}, z_{k':}\rangle z_{\ell b}^2\right] = 0 \tag{52}$$

Finally for the feature $\mathcal{H}(Z)_{\ell b} = m_2 z_{\ell b}$ we have on the main diagonal $\mathbb{E}[m_2^2 z_{\ell b}^2] = m_2^2$ and $0$ everywhere else.

Therefore we've concluded that $\Lambda(Z)$ is a block diagonal matrix because the $\ell \neq \ell'$ blocks are near zero. All that remains is to verify that the diagonal blocks are full rank.

1. Pairs to Pairs: $m_2^3 I$ is full rank with min eigenvalue $m_2^3$

2. Singles to Singles: $(m_4 - m_2^2)m_2 I$ is full rank with min eigenvalue $(m_4 - m_2^2)m_2$.

$\qquad\qquad\qquad\qquad\qquad\qquad\qquad\qquad\qquad\qquad\qquad\qquad\qquad\qquad\qquad\qquad\quad\square$

Finally we provide a simple error bound for approximate empirical risk minimizers to demonstrate the robustness of the conclusions in Lemma A.3.

**Lemma A.7** (Identifiability with Error). *Let $\Omega_{\epsilon-ERM}$ be the set of $\epsilon$-approximate empirical risk minimizers,*

$$\Omega_{\epsilon-ERM} =$$

$$\left\{ \Theta \in \Omega_H \mid \mathbb{E}_{(Z_i, y_i) \in D}\left[(MHLA_\Theta(Z_i) - y_i)^2\right] \leq \epsilon \right\}.$$

*Then we have for any $\Theta, \Theta' \in \Omega_{\epsilon-ERM}$ that for all inputs $Z \in \mathbb{R}^{d \times n}$*

$$\|MHLA_\Theta(Z) - MHLA_{\Theta'}(Z)\| \leq \frac{\epsilon}{\lambda_{\min}(\Lambda_D)}\|Z\|_F^6. \tag{22}$$

*Proof.*

$$\|\mathrm{MHLA}_\Theta(Z) - \mathrm{MHLA}_{\Theta'}(Z)\|^2 = \sum_{a \in [d]} \left(\langle p_a(\Theta) - p_a(\Theta'), \mathcal{H}(Z)\rangle\right)^2$$

$$\leq \sum_{a \in [d]} \|p_a(\Theta) - p_a(\Theta')\|^2 \|\mathcal{H}(Z)\|^2$$

$$\leq \left(\sum_{a \in [d]} \|p_a(\Theta) - p_a(\Theta')\|^2\right) \|Z\|_F^6$$

$$\leq \frac{\epsilon}{\lambda_{min}(\Lambda_D)}\|Z\|_F^6 \tag{53}$$

Here the first equality follows from the linearization exhibited in Lemma E.1. The first inequality is cauchy schwarz. In the second inequality we apply a crude upper bound that no more than 6'th degree polynomials that are products of three squares of entries in $Z$ are involved in $\|\mathcal{H}(Z)\|^2$.

$$\|\mathcal{H}(Z)\|^2 \leq \sum_{a,a',a'' \in [d],\, b,b',b'' \in [n]} Z_{ab}^2 Z_{a'b'}^2 Z_{a''b''}^2 \leq \|Z\|_F^6 \tag{54}$$

The last inequality comes from the fact that $\Theta, \Theta'$ are $\epsilon$ approximate empirical risk minimizers. Therefore we know

$$\lambda_{min}(\Lambda_D) \sum_{a \in [d]} \|p_a(\Theta) - p_a(\Theta')\|^2 \leq \sum_{a \in [d]} \left(\langle p_a(\Theta) - p_a(\Theta'), \mathcal{H}(Z)\rangle\right)^2 \leq \epsilon \tag{55}$$

which implies

$$\sum_{a \in [d]} \|p_a(\Theta) - p_a(\Theta')\|^2 \leq \frac{\epsilon}{\lambda_{min}(\Lambda_D)} \tag{56}$$

which concludes the proof. $\qquad\square$

**Lemma E.2** (Identifiability with Error and Noise in Realizability). *Let $D = \{(Z_i, y_i)\}_{i \in [N]}$ be a dataset such that $y_i = MHLA(Z_i) + \zeta_i$ for $\zeta_i$ i.i.d and bounded. Let $\Omega_{\epsilon-ERM}$ be the set of $\epsilon$-approximate empirical risk minimizers.*

$$\Omega_{\epsilon-ERM} = \left\{ \Theta \in \Omega_H \mid \mathbb{E}_{(Z_i, y_i) \in D}\left[ (MHLA_\Theta(Z_i) - y_i)^2 \right] \leq \epsilon \right\}. \tag{57}$$

*Let $\max_{i \in [N]} \|Z_i\|_F \leq B$. Then we have for any $\Theta, \Theta' \in \Omega_{\epsilon-ERM}$ that for all inputs $Z \in \mathbb{R}^{d \times n}$*

$$\|MHLA_\Theta(Z) - MHLA_{\Theta'}(Z)\| \leq \frac{\epsilon - \frac{1}{N}\sum_{i \in [N]} \zeta_i^2 + \frac{B^2}{N} \log(\delta^{-1})}{\lambda_{\min}(\Lambda_D)} \|Z\|_F^6. \tag{58}$$

*Proof.* The proof follows directly from Lemma A.7 but we incorporate the $\zeta_i$ terms as is standard in analyses of linear regression. $\qquad\square$

# F  Programs Expressible as Fixed Depth Linear Transformer

In this section we build out examples of programs that can be expressed as fixed depth linear transformers. Expressibility results can be carried out in a variety of equivalent ways. The main takeaway, is that the computation history of TM $M$ on word $x$, when written down "step by step" can be captured by next token prediction of linear attention. This is because the key-query-value naturally implements a table lookup sometimes referred to as "associative memory" or "in context linear regression" in the linear case.

The notion of an Autoregressive MHLA Program is useful for condensing the proofs of expressibility. We write such programs in an object oriented syntax with each token representing an object with multiple attributes. Attributes can be updated and looked up from other objects using a generalized lookup akin to associative memory.

---

**Algorithm 3** Autoregressive MHLA Program

---

1: Instantiate N instances OBJ = $\{\text{obj}(i)\}_{i \in [N]}$ of Class with set of Attributes $\{\text{Attr}_1, \text{Attr}_2, ..., \text{Attr}_k\}$
2: Each Attribute takes on values in an alphabet $\Sigma_{\text{Attribute}}$
3: **for** iter $\in$ [T] **do**
4:     Let obj[$r$] be the rightmost token
5:     Let obj[$r + 1$] be a new token initialized with positional embedding obj[$r + 1$].pos = $r + 1$
6:     **for** each {AttrSource, AttrDest} in {Pairs of Attributes in Class} **do**
7:         #AttrKey and AttrValue can be any pair of Attributes (and can be distinct from VarSource/VarDest)
8:         LookupDict = {{obj.AttrKey: obj.AttrValue} for obj in OBJ}
9:         # if multiple objects have same obj.AttrKey then returns sum of obj.AttrValues which we aim to avoid
10:        Let $\mathcal{B}_Q$ be any function from $\Sigma_{\text{AttrSource}}$ to $\Sigma_{\text{AttrKey}}$
11:       Let $\mathcal{B}_V$ be any function from $\Sigma_{\text{AttrValue}}$ to $\Sigma_{\text{AttrDest}}$
12:       Let query = $\mathcal{B}_Q(\text{obj}[r].\text{AttrSource})$
13:       **if** query in LookupDict.Keys **then**
14:         obj[r+1].AttrDest = $\mathcal{B}_V(\text{LookupDict}(\text{query}))$
15:       **end if**
16:     **end for**
17:     Append next token OBJ = $\{\text{obj}[i]\}_{i \in [r]} \cup \{\text{obj}[r + 1]\}$
18:     $r = r + 1$
19: **end for**

---

**Lemma F.1.** *For any program $\mathcal{P}$ written in the form of algorithm 6, there exists corresponding MHLA parameters $\Theta \in \Omega_H$ such that $\text{MHLA}_\Theta(Z) = \mathcal{P}(Z)$.*

*Proof.* We set some matrices to implement lookup tables. For any function of $f : A \to B$ for sets $A$ and $B$ there is a canonical representation of the input domain as orthogonal unit vector $v_1, v_2, ..., v_{|A|} \in \mathbb{R}^A$ and output domain as another set of orthogonal unit vectors $u_1, u_2, ..., u_{|B|} \in \mathbb{R}^B$. Therefore, there is a matrix $G_f$ that maps input vectors to output vectors satisfying $G_f v_i = u_j$ for $j = f(i)$ for all $i \in [A]$ and $j \in [B]$.

For functions $f : \Sigma_{\text{AttrSource}} \to \Sigma_{\text{AttrKey}}$ and $f' : \Sigma_{\text{AttrValue}} \to \Sigma_{\text{AttrDest}}$ we associate matrices $B_Q \in \mathbb{R}^{|\Sigma_{\text{AttrSource}}| \times |\Sigma_{\text{AttrKey}}|}$ and $B_V \in \mathbb{R}^{|\Sigma_{\text{AttrValue}}| \times |\Sigma_{\text{AttrDest}}|}$ respectively.

Then we form $\{V_h, Q_h\}_{h \in [H]}$ as follows. Let $V$ be the matrix that is all zeros with $B_V$ in the rows associated with $\Sigma_{\text{AttrSource}}$ and the columns associated with $\Sigma_{\text{AttrKey}}$. Let $Q$ be the matrix that is all zeros with $B_V$ in the rows associated with $\Sigma_{\text{AttrValue}}$ and the columns associated with $\Sigma_{\text{AttrDest}}$.

In each layer we have multiple heads, each one performs the lookup operation for each pair of attributes in the class. $\qquad\square$

## F.1  Construction of UTM

Now we proceed with our construction of an Autoregressive MHLA-Program for UTM. The UTM requires a small number of operations captured by an Autoregressive MHLA-Program.

We define an embedding function that takes as input a TM $M$ and word $x$ such that

**Definition F.2** (Embedding). Let $M$ be a TM over state space $Q$, alphabet $A$, transition function $\delta$. Then

$$\text{Embedding}(M) = \begin{bmatrix} q_0 & q_1 & \cdots & q_k & \# \\ a_0 & a_0 & \cdots & a_0 & \# \\ \delta(q_0, a_0) & \delta(q_1, a_0) & \cdots & \delta(q_k, a_0) & \# \\ a_1 & a_1 & \cdots & a_1 & \# \\ \delta(q_0, a_1) & \delta(q_1, a_1) & \cdots & \delta(q_k, a_1) & \# \end{bmatrix} \tag{59}$$

Let $p_1, p_2, ..., p_\delta$ be "positional encodings" that assign unique id's for every letter in the word $x$.

$$\text{Embedding}(x) = \begin{bmatrix} p_1 & p_2 & \cdots & p_i & p_{i+1} & \cdots & p_\delta & \# \\ x_1 & x_2 & \cdots & x_i & x_{i+1} & \cdots & x_\delta & \# \\ 0 & 0 & \cdots & q & 0 & \cdots & 0 & \# \end{bmatrix} \tag{60}$$

Then we define Embedding(M,x) to be

$$\text{Embedding(M,x)} = \begin{bmatrix} \text{Embedding}(M) & 0 \\ 0 & \text{Embedding}(x) \end{bmatrix} \tag{61}$$

Henceforth we will write the construction in the syntax of an Autoregressive MHLA-Program instead of matrices with blocks of zeros and token embeddings to save space.

**Lemma B.1** (UTM Expressibility). *Let $\Delta(\hat{\mathcal{Q}}, \hat{\Sigma}, \hat{n}, \hat{\Phi})$ be the set of Turing machines $M = \{\delta, \Sigma, \mathcal{Q}, q_{start}, q_{accept}, q_{reject}\}$ and words $x \in \Sigma^*$ with number of states, size of alphabet, size of input, and number of steps in computation history bounded by $\hat{\mathcal{Q}}, \hat{\Sigma}, \hat{n}, \hat{\Phi}$ respectively. For any $(M, x) \in \Delta$, let $\{x_t\}_{t \in [\Phi]}$ be the computation history of the UTM on $(M, x)$. Let the autoregressive computation history (see Definition C.2) of MHLA$_\Theta$ on input $(M, x)$ be denoted $CH_\Theta(M, x) = \{Z^1, Z^2, ..., Z^\Phi\}$. Then there exists a set of parameters $\Theta \in \Omega_H$ for $H = O(\hat{n}\hat{\Phi}\hat{\Sigma})$ and embedding dimension $d = O(\hat{n}\hat{\Phi}\hat{\Sigma} \max(\hat{\Sigma}, \hat{\mathcal{Q}}))$, such that for all $(M, x) \in \Delta$, the TM computation history at time step $t$ is equivalent to the autoregressive computation history at time step $c(t)$ where $c(t) \leq O((n + t)t)$ i.e $Z^{c(t)}[: -length(x^t))] = x^t$. Furthermore, this can be achieved with 2 bits of precision.*

The construction is given in the language of Autoregressive MHLA-Programs in algorithm 6 which provides the instruction set for writing the next letter in the computation history onto the output tape.

*Proof.* **Proof Idea:** A few elementary operations can be captured by a MHLA-program which can be composed to output the computation history of $M$ on $x$. We begin by introducing some notation for the "Lookup" operation which we build into copy, move, and if-then which are all the operations required to construct the UTM.

**General Lookup:** For each lookup there are three objects that are involved. Let Token= obj$[r]$ be the "source" which is always the rightmost token. An attribute from the source object known as AttrSource is linearly transformed to form a "query". Lookup involves a table $T = \{\text{obj[i].AttrKey: obj[i].AttrValue}\}_{i \in [r]}$ which is used to match an AttrKey to look up an AttrValue from an object obj$[p]$ that we denote the "target". Note, that if the obj[i] has an AttrKey that is zero, it is the same as not being in the table. In the pseudocode algorithm 6 these zero attributes are denoted as "None".

Given a query, we copy the associated AttrValue from the lookup table $T$ and update AttrDest in an object NextToken= obj$[r + 1]$ which we denote the "destination". Multiple lookup operations can be performed in parallel by many heads with each head responsible for a single lookup.

To output each letter of the computation history, we increase the number of tokens $r$ by a constant $c$. We refer to the set of contiguous tokens $[0, c], [c, 2c], etc.$ involved in the computation of a single letter as a "block". Here block[i] = $\{\text{obj}[j]\}_{j \in [ic, (i+1)c]}$. We construct a different set of heads to act on each token and enforce that the nonzero rows that each block of tokens occupy are disjoint. Furthermore, within a block, the states of each token occupies a disjoint set of rows except when they are used to construct a table. Tables are the only case where we want tokens to occupy the same rows. In this manner the following abstraction can be made.

At the beginning of each block starting with obj[r], we can lookup attributes from anywhere in OBJ that we want to load into different attributes in obj[r]. Then we can apply any sequence of if-then

statements involving the attributes of obj[r] to update the attributes (or create new attributes). To run the UTM we need a few simple primitives denoted Lookup and If-Then.

**Construction of Primitives:** We write down the construction by constructing a sufficient set of primitives Lookup and If-Then. We also include Copy which is a special case of Lookup that is used frequently.

**Lookup:** When the transforms $B_Q$ and $B_V$ are the identity we denote the lookup operation for table $T$ where we query an attribute $s'$ of obj$[r]$ to update the attribute $s$ of obj[r+1] as obj[r+1].s = Lookup(T,obj[r].s')

**Copy:** A special case of lookup is copy, where we need to copy attributes from tokens that are at an offset $-k$ for $k \in [r]$. This can be done by setting $\mathcal{B}_Q$ to permute the positional encoding by $-k$ positions. Then the query matches the key that is the positional encoding of the target object. Let $s, s'$ be target and destination attributes. We denote the copy operation of the attribute $s'$ of the obj at offset $-k$ from $r$ into the attribute $s$ of the destination object to be obj[r+1].s = Copy(obj[r-k].s').

**If-Then:** We write down an If-Then Program algorithm 4 and a corresponding Autoregressive MHLA-Program algorithm 5 to implement If-Then. An If-Then program looks up whether an attribute $x$ is equal to any of attributes $a_1, a_2, ..., a_k$ then we set attribute $x'$ to $b_1, b_2, ..., b_k$ respectively. This is achieved by copying the attributes $a_i$ and $b_i$ into dummy attributes $s0$ and $s1$ for all $i$ in $k$ for a series of $k$ consecutive tokens. This creates a table with key $s0$ and value $s1$. Then we use attribute $x$ as the query, which looks up the corresponding value $s1$ which we use to update an attribute $x'$.

---

**Algorithm 4** If-Then Program

1: # If attribute x is equal to any of $a_1, a_2, ..., a_k$ then set attribute $x'$ to $b_1, b_2, ..., b_k$ respectively
2: **if** Token.x == Token.$a_1$: **then**
3:     NextToken.x' = Token.$b_1$
4: **end if**
5: **if** Token.x == Token.$a_2$: **then**
6:     NextToken.x' = Token.$b_2$
7: **end if**
8: . . .
9: **if** Token.x == Token.$a_k$: **then**
10:     NextToken.x' = Token.$b_k$
11: **end if**

---

**Algorithm 5** MHLA If-Then Program

1: # If attribute x is equal to any of $a_1, a_2, ..., a_k$ then set attribute $x'$ to $b_1, b_2, ..., b_k$ respectively
2: token[r+1].s0 = token[r].$a_1$
3: token[r+1].s1 = token[r].$b_1$
4: NEXT TOKEN $r = r + 1$
5: token[r+1].s0 = token[r].$a_2$
6: token[r+1].s1 = token[r].$b_2$
7: . . .
8: NEXT TOKEN $r = r + 1$
9: token[r+1].s0 = token[r].$a_k$
10: token[r+1].s1 = token[r].$b_k$
11: NEXT TOKEN $r = r + 1$
12: Table T = $\{$obj[i].s0 : obj[i].s1$\}_{i \in [r, r-k+1]}$
13: token[r+1].x' = Lookup(T,token[r].x)

---

$\square$

## F.2 Proofs For Learning UTM

**Lemma C.5** (Learning a UTM). *Let* $\Theta \in \Omega_H$ *in dimension* $d$ *be the MHLA parameters in Lemma B.1. Let* $\{M_i, x_i\}_{i \in [N]}$ *be pairs of TM's* $M$ *and words* $x$ *of maximum length* $n$ *drawn*

---

**Algorithm 6** Simplified Instruction Set MHLA Program for UTM for a single block

---

1: # Initialize Lookup Tables for TM M and tape $T_1$
2: # $\delta(q, a) = $ [next-state, next-letter, next-move]
3: M = $\{q : [a_0, \delta(q, a_0), a_1, \delta(q, a_1)]\}_{q \in Q}$
4: $T_1 = \{\text{token[i].PosEncoding: token[i].Letter}\}_{i \in [r]}$
5: # Begin Loading Information from M and previous tokens on tape
6: # First copy letter/state from token -N-1 positions away
7: # Attribute s(-1) = {letter, state} where state can be equal to None
8: NextToken.s(-1) = Copy(Token[-N-1].s0)
9: # Second copy letter/state from token -N positions away
10: # Attribute s0 = {letter, state} where state can be equal to None
11: NextToken.s0 = Copy(Token[-N].s0)
12: # Third copy letter/state from token -N+1 positions away
13: # Attribute s1 = {letter, state} where state can be equal to None
14: NextToken.s1 = Copy(Token[-N+1].s0)
15: NEXT TOKEN $r = r + 1$
16: #Split into three branches to handle left, head, and right positions relative to head
17: **RUN BRANCH 1** (Token is Left of Head Position) See algorithm 7
18: **RUN BRANCH 2** (Token is at Head Position) See algorithm 7
19: **RUN BRANCH 3** (Token is Right of Head Position) See algorithm 7

---

*i.i.d. from a distribution $\mathcal{D}$. Let $Z_i = Embed(M_i, x_i)$. For each TM/word pair $(M_i, x_i)$ let $CH_\Theta(Z_i) = \{Z_i^1, Z_i^2, ..., Z_i^\Phi\}$ be the $\Phi$-step autoregressive computation history of MHLA$_\Theta$ on $Z_i$. Let D be the dataset $D := \{(CH_\Theta(Z_i)^t, y_i^{t+1})\}_{i \in [N], t \in [T]}$ where $y_i^{t+1} = MHLA_\Theta(Z_i^t)$. Then Algorithm 1 applied to input D returns $\hat{\Theta} \in \Omega_H$ for $H \leq d^2$ such that with probability $1 - \delta$*

$$\mathbb{E}_{(Z,y) \in \mathcal{D}}\left[\left(MHLA_{\hat{\Theta}}(Z) - y\right)^2\right] \leq \epsilon \tag{26}$$

*for sample complexity $N = poly(d, \epsilon^{-1}, \log(\delta^{-1}))$. Then with probability $1 - \delta$ over the randomness in the data, the probability over $\mathcal{D}$ that the $\Phi$-step autoregressive computation history $CH_{\hat{\Theta}}(M, x)$ and $CH_\Theta(M, x)$ differ is upper bounded by*

$$\Pr_{(M,x) \sim \mathcal{D}}[CH_{\hat{\Theta}}(M, x) \neq CH_\Theta(M, x)] \leq O(\epsilon\Phi). \tag{27}$$

**Corollary F.3.** *In particular, for sample complexity $N = poly(d, \epsilon^{-1}, \log(\delta^{-1}), n, t)$, by Lemma B.1, we have with probability $1 - \delta$ over the randomness in the data that the probability that the $c(t)$ step of the computation history of MHLA$_{\hat{\Theta}}$ is equal to $x_t$ is*

$$\Pr_{(M,x) \sim \mathcal{D}}\left[CH_{\hat{\Theta}}(M, x)^{c(t)}[: -k_t] = x^t\right] \geq 1 - \epsilon, \tag{62}$$

*where $c(t) \leq O((n + t)t)$. That is, the computation history of the MHLA returned by algorithm 1 is equal to the computation history of M on x.*

*Proof.* We have from Theorem 2.2 that algorithm 1 returns $\hat{\Theta}$ such that

$$\mathbb{E}_{(Z,y) \in \mathcal{D}}\left[\left(MHLA_{\hat{\Theta}}(Z) - y\right)^2\right] - \min_{\Theta \in \Omega_H} \mathbb{E}_{(Z,y) \in \mathcal{D}}\left[\left(MHLA_\Theta(Z) - y\right)^2\right] \leq \epsilon \tag{63}$$

Then to obtain an error bound on the $\Phi$ step computation history, which involves $O(n\Phi)$ tokens, we just observe that by union bound each step rounds to an incorrect set of tokens with probability less than $\epsilon$. Therefore, over $O(\Phi)$ steps the error probability is upper bounded by $\epsilon\Phi$. Equivalently

$$\Pr_{(M,x) \sim \mathcal{D}}[CH_{\hat{\Theta}}(M, x) \neq CH_\Theta(M, x)] \leq O(\epsilon\Phi). \tag{64}$$

Then proving Corollary F.3 is a simple exercise. For a larger sample complexity $N = poly(d, \epsilon^{-1}, \log(\delta^{-1}), n, t)$, by Lemma B.1, we have that the probability that every token of the autoregressive computation history of MHLA$_{\hat{\Theta}}$ is equal to $x_t$ is

$$\Pr_{(M,x) \sim \mathcal{D}}\left[CH_{\hat{\Theta}}(M, x)^{c(t)}[: -k_t] = x^t\right] \geq 1 - \epsilon \tag{65}$$

$\square$

**Algorithm 7** Branches to handle cases Left of Head, Head, and Right of Head

1: #Split into three branches to handle left, head, and right positions relative to head
2: **BRANCH 1** (Token is Left of Head Position)
3: # we have loaded a state q into s1 (if left of head) and next we load $[a_0, \delta(q, a_0), a_1, \delta(q, a_1)]$ into s2
4: NextToken.s2 = Lookup(M,Token.s1.state)
5: NEXT TOKEN $r = r + 3$
6: **if** Token.s2.letter == $a_0$ **then**
7:     NextToken.s3 = $\delta(q, a_0)$ = [q',w',L/R]
8: **end if**
9: **if** Token.s2.letter == $a_1$ **then**
10:     NextToken.s3 = $\delta(q, a_1)$ = [q',w',L/R]
11: **end if**
12: NEXT TOKEN r = r+3
13: **if** Token.s3.move == L **then**
14:     NextToken.return-letter = Token.s0.letter
15:     NextToken.return-state = q'
16: **end if**
17: **if** Token.s3.move == L **then**
18:     NextToken.return-letter = Token.s0.letter
19:     NextToken.return-state = None
20: **end if**
21: **BRANCH 2** (Token is at Head Position)
22: # we have loaded a state q into s0 and next we load $[a_0, \delta(q, a_0), a_1, \delta(q, a_1)]$ into s2
23: NextToken.s2 = Lookup(M,Token.s0.state)
24: NEXT TOKEN r = r+3
25: **if** Token.s2.letter == $a_0$ **then**
26:     NextToken.s3 = $\delta(q, a_0)$ = [q',w',L/R]
27: **end if**
28: **if** Token.s2.letter == $a_1$ **then**
29:     NextToken.s3 = $\delta(q, a_1)$ = [q',w',L/R]
30: **end if**
31: NEXT TOKEN r = r+3
32: **if** Token.s3.next-letter is not None **then**
33:     NextToken.return-letter = Token.s3.next-letter
34:     NextToken.return-state = None
35: **end if**
36: **BRANCH 3** (Token is Right of Head Position)
37: # we have loaded a state q into s(-1) and next we load $[a_0, \delta(q, a_0), a_1, \delta(q, a_1)]$ into s2
38: NextToken.s2 = Lookup(M,Token.s(-1).state)
39: NEXT TOKEN r = r+3
40: **if** Token.s2.letter == $a_0$ **then**
41:     NextToken.s3 = $\delta(q, a_0)$ = [q',w',L/R]
42: **end if**
43: **if** Token.s2.letter == $a_1$ **then**
44:     NextToken.s3 = $\delta(q, a_1)$ = [q',w',L/R]
45: **end if**
46: NEXT TOKEN r = r+3
47: **if** Token.s3.move == L **then**
48:     NextToken.return-letter = Token.s0.letter
49:     NextToken.return-state = None
50: **end if**
51: **if** Token.s3.move == R **then**
52:     NextToken.return-letter = Token.s0.letter
53:     NextToken.return-state = Token.s3.next-state
54: **end if**

**Lemma A.8** (Learning UTM from Certifiably Identifiable Data). *Let $D = \{(Z_i, y_i)\}_{i \in [N]}$ be a dataset satisfying $y_i = MHLA_\Theta$ for $\Theta \in \Omega_H$ being the expressibility parameters of Lemma B.1 for the set of TM's/words $(M, x) \in \Delta(\hat{\mathcal{Q}}, \hat{\Sigma}, \hat{n}, \hat{\Phi})$. If $D$ is certifiably identifiable with $\lambda_{min}(\Lambda_D) > \eta$, then there is a poly$(d, N, \hat{Q}, \hat{\Sigma}, \hat{n}, \hat{\Phi}, \eta^{-1})$ time algorithm that outputs a set of parameters $\hat{\Theta} \in \Omega_{d^2}$ such that for all TM's $M$ and input words $x$ in $\Delta(\hat{\mathcal{Q}}, \hat{\Sigma}, \hat{n}, \hat{\Phi})$, we have*

$$CH_{\hat{\Theta}}(M, x)^{c(t)}[:-k_t] = x^t . \tag{23}$$

*The $c(t)$ step of the autoregressive computation history of $\hat{\Theta}$ is equal to the $t$'th step of the computation history of $M$ on $x$.*

*Proof.* The proof follows from the quantitative version of Lemma A.7. Using the given that $\lambda_{min}(\Lambda_D) > \eta$, we conclude that for any $\hat{\Theta} \in \Omega_{\epsilon-\text{ERM}}$ that for all inputs $Z \in \mathbb{R}^{d \times n}$

$$\|\text{MHLA}_{\hat{\Theta}}(Z) - \text{MHLA}_\Theta(Z)\| \leq \frac{\epsilon}{\eta}\|Z\|_F^6. \tag{66}$$

If we select a sufficiently small $\epsilon = 1/\text{poly}(d, N, |Q|, |\Sigma|, n, t, \eta^{-1})$ then we can ensure

$$\Pr_{(M,x) \sim \mathcal{D}} \left[\text{CH}_{\hat{\Theta}}(M, x)^{c(t)}[:-k_t] = x^t\right] \geq 1 - \epsilon \tag{67}$$

.

The runtime then scales with poly$(d, N, |Q|, |\Sigma|, n, t, \eta^{-1})$ as desired. $\qquad\square$

# G  Additional Definitions

**Definition G.1** (Orthogonal Embeddings). Let Embed be a function Embed $: \Sigma \to \mathbb{R}^{|\Sigma|}$. Let $\Sigma$ be an alphabet and let $e_1, e_2, ..., e_{|\Sigma|} \in \mathbb{R}^{|\Sigma|}$ be a basis of orthogonal unit vectors. Then for each letter $a$ in an alphabet $\Sigma$, we define $\text{Embed}(a) = e_a$ where we associate a different unit vector to each letter.

We adopt a naive "rounding" scheme for converting vectors into tokens. This can be done in a variety of ways, and we choose to simply round the entries of the vector embeddings to the nearest token embedding.

**Definition G.2** (Rounding). For any vector $v = (v_1, v_2, ..., v_d) \in \mathbb{R}^d$, let $\text{Round}(v) = e_j$ for $j = \arg\max_{i \in [d]} \langle v, e_i \rangle$. Since we use orthogonal unit vectors for token embeddings we will refer to $\text{Round}(v)$ as a token. We will often refer to a matrix $Z \in \mathbb{R}^{d \times n}$ as being equivalent to a series of $n$ tokens $a_1, a_2, ..., a_n$ to mean $\text{Round}(Z[:, i]) = a_i$ for all $i \in [n]$.

---

**Algorithm 8** Extract Features

---

1: **Input:** Data $D := \{Z_i\}_{i \in [N]}$ for $Z_i \in \mathbb{R}^{d \times n_i}$ and $y_i \in \mathbb{R}^d$
2: **for** $Z_i \in D$ **do**
3:     Let $z_1, z_2, ...z_d$ be the rows of $Z_i$ and let $z_{a,b}$ be the $(a, b)$ entry of $Z_i$
4:     **for** $j \in [d]$ **do**
5:         **for** $k \in [d]$ **do**
6:             **for** $\ell \in [d]$ **do**
7:                 Let $\mathcal{X}_i \in \mathbb{R}^{d \times d^2}$ be defined as follows
8:                 $\mathcal{X}_i[j, kd + \ell] = [\langle z_{j:}, z_{k:} \rangle z_{\ell n_i}]$
9:             **end for**
10:         **end for**
11:     **end for**
12: **end for**
13: **Return:** $\{\mathcal{X}_i\}_{i \in [N]}$ such that

$$\mathcal{X}_i := \begin{bmatrix} \langle z_{1:}, z_{1:} \rangle z_{1n_i} & \langle z_{1:}, z_{2:} \rangle z_{1n_i} & \cdots & \langle z_{1:}, z_{d:} \rangle z_{1n_i} & \cdots & \langle z_{1:}, z_{d:} \rangle z_{dn_i} \\ \langle z_{2:}, z_{1:} \rangle z_{1n_i} & \langle z_{2:}, z_{2:} \rangle z_{1n_i} & \cdots & \langle z_{2:}, z_{d:} \rangle z_{1n_i} & \cdots & \langle z_{2:}, z_{d:} \rangle z_{dn_i} \\ \vdots & \vdots & \ddots & \vdots & \ddots & \vdots \\ \langle z_{d:}, z_{1:} \rangle z_{1n_i} & \langle z_{d:}, z_{2:} \rangle z_{1n_i} & \cdots & \langle z_{d:}, z_{d:} \rangle z_{1n_i} & \cdots & \langle z_{d:}, z_{d:} \rangle z_{dn_i} \end{bmatrix}. \quad (68)$$

---

## G.1  Training details of attention networks

We use Adam Kingma and Ba [2014] optimizer to train linear attention model Equation (4) and the full Transformer Vaswani et al. [2017] models.

| hyper parameter | search space |
|---|---|
| $d$ input dimension | [2, 4, 8, 16] |
| $m$ number of heads | [1, 2, 4, 8, 16] |
| $n$ number of layers | [1, 2, 4] |
| learning rate | [0.01, 0.001] |
| batch size | [32, 64] |
| optimizer | AdamW Loshchilov and Hutter [2018] |

## G.2  Training details in DFA Execution

We use the Llama variant of the Transformer arhitecture from Touvron et al. [2023]. We run each setting with $N$ number of training examples with the following different values $N \in \{16, 32, 64, 128, 256, 512, 1024, 2048, 4096, 6144, 8192, 12290, 16384, 20480, 32768, 65536\}$. The other hyper parameters are given in the below table.

| hyper parameter | search space |
| --- | --- |
| $d$ input dimension | [2048] |
| $m$ number of heads | [16] |
| $n$ number of layers | [4] |
| learning rate | [0.00025] |
| epochs | 100 |
| optimizer | AdamW Loshchilov and Hutter [2018] |

## G.3  Additional Experiments

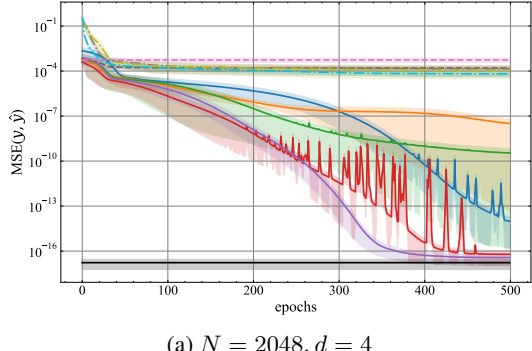

(a) $N = 2048, d = 4$

Figure 3: **Performance comparison of multi-head, multi-layer linear attention models and the original Transformer model (denoted as *full*).** We trained using SGD on synthetic data generated from a single-layer linear attention model for varying training set sizes ($N$) and input dimensions ($d$), number of heads $m$, and number of layers $n$. We present mean squared error of the predictions w.r.t number of training epochs. Results demonstrate that multi-head architectures converge faster on different input dimensions and match the performance of our algorithm 1 (convex algorithm). Increasing the number of layers or incorporating multilayer perceptrons (MLPs) and layer normalization did not yield consistent improvements. Shading indicates the standard error over three different runs.

## G.4  Learning the Computation History of Deterministic Finite Automata

Universal automata (like the universal Turing machine discussed in Appendix F.2) receive descriptions of other automata as input, and simulate them to produce an output. Here we empirically evaluate the ability of MHLA models to perform universal simulation of deterministic finite automata (DFAs). We limit our study to DFAs with a maximum number of states ($N$), alphabet size ($V$), and input length ($L$). While recent work on in-context learning [Akyürek et al., 2024] has focused on inferring DFA behavior from input–output examples, here, we aim to simulate DFAs given explicit descriptions of their state transitions as input—a task somewhat analogous to *instruction following* in large scale language models.

The construction in Lemma C.5 shows that a linear attention layer can output the polynomially bounded computation history of any TM (and therefore any DFA). Our construction requires embedding size linear with maximum length of computation history, number of states and alphabet size. Therefore, we predict the data requirements are polynomial in each of $N, V$ and $L$.

**Dataset**  Our dataset consists of strings containing three components: the input DFA's transition function $\delta : \mathcal{Q} \times \Sigma \to \mathcal{Q}$, the input word $x \in \Sigma^L$ and the computation history $h \in \mathcal{Q}^L$ which is the sequence of states visited in the DFA as it decides if $x$ is in its language. The first two components are the input to the model, while the computation history is the target output. We adopt the following schema for representing $\delta, x$, and $h$:

$$\underbrace{(s_i, w, s_j), \ldots, \forall_{s_i \in \mathcal{Q}, w \in \Sigma} \in \delta}_{\text{DFA transition function}} \mid \underbrace{w_0 w_1 \ldots w_L}_{\text{word}} \mid \underbrace{(s^0 w_0 s^1), (s^1 w_1 s^2), \ldots, (s^{L-1} w_L s^L)}_{\text{computation history}}$$

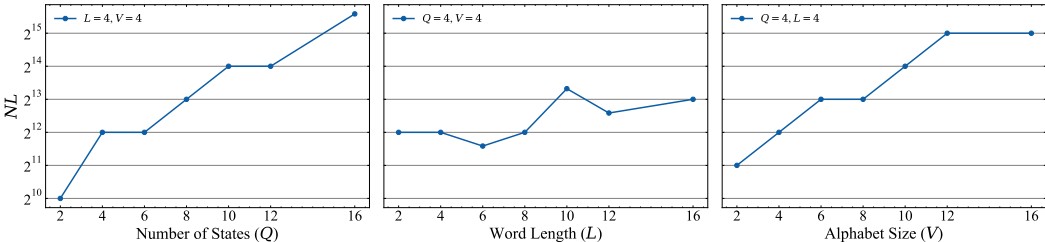

Figure 4: **Data requirement for universal DFA simulation:** We train a fixed sized Transformer (4-layers, 16 heads and 2048 hidden dimensions) to simulate a DFA given a transition table and input word. The vertical axis shows the number of tokens (expressed as word length $L$ times the number of examples $Q$) required to obtain 99% next token accuracy.

We encode each input-output relation in the transition function as a sequence of three tokens $(s_i, w, s_j)$ where $\delta(s_i, w) = s_j$. We also include two parantheses to separate each triplet of tokens for a total of five tokens for each input-output relation. The total description length of $\delta$ is then $5Q\Sigma$. We encode word $x$ of length $L$ as a sequence of $L$ tokens. Finally, we encode the computation history as the sequence of state transitions the DFA visits when deciding if $x$ is in its language. Here we designate $s0$ as the start state, and let $s^i = \delta(s^{i-1}, w^{i-1})$. Each state transition is again represented by a triplet $(s, w, \delta(s, w))$. We train an autoregressive Transformer model using cross-entropy loss to predict the computation history tokens given the transition function and word. Please refer to Appendix G.2 for hyperparameter details.

**Results**   In Figure 4, we vary each of the parameters $Q$, $L$ and $V$, while the other two parameters are fixed to a constant (in this case we fix them to be 4). Then, on the vertical axis, we display the minimum number of tokens (number of examples times the word length) required to get 99% accuracy on the next token prediction. Plots are suggestive of a sub-exponential dependence on DFA complexity.

