# OpenReview forum: "Learning Linear Attention in Polynomial Time"
_NeurIPS.cc/2025/Conference — NeurIPS 2025 oral_

### Official Review · Reviewer_EkNE · 2025-07-03

**Clarity:** 3
**Significance:** 4
**Originality:** 4
**Rating:** 6
**Confidence:** 4

**Summary:**

The paper establishes the strong, agnostic PAC-learnability of a single layer of multi-head linear attention, for regression tasks.  This is accomplished by first transforming the function into an elementwise product of (1) a matrix derived from the input and (2) a matrix derived from the parameters.  This means that optimizing over H heads equates to optimizing over the set of rank-H matrices.  This leads to a result that the training-set optimal matrix can be identified by effectively fitting a linear regression and applying singular value decomposition. The next result shows that the minimizer is identifiable (surprising in my subjective opinion) if the dataset is realizable by MLHA or there are enough heads in the model.  Together, these results give a poly-time procedure for learning any circuit family that MHLA can handle when the training data satisfy that condition.  The procedure recovers a Turing machine that the learned MHLA will simulate on any input within a size budget.

Experiments explore overparameterization by adding heads -- which leads to a convex problem in the theoretical setup -- gives optimization benefits (more so than adding layers).  Further experiments empirically verify the identifiability certificate in various settings.

**Questions:**

Line 60:  polynomial time *in what*?  I suspected at first that there's a dependency on the number of training instances, and this should be presented honestly up front, not buried later in the paper.  Later, if I understood correctly, there was not (it seems it's about the dimensionality of the key/query vectors ...).

**Ethical Concerns:**

["NO or VERY MINOR ethics concerns only"]

**Final Justification:**

Thanks for the response.

**Limitations:**

Yes

**Quality:**

4

**Strengths And Weaknesses:**

Strengths

The paper presents incredibly important theoretical advances in what (some key parts of an important variant of) the transformer network can learn.  The ideas are clearly conveyed and bring together tools from a wide range of CS and machine learning.  This is creative work and an important step toward a deeper understanding of an incredibly influential class of architectures.

The paper also does a nice job of situating the results in the context of extensive related work.

The experiments are thoughtfully selected and well chosen, e.g., connecting the theoretically derived algorithm with behavior of SGD, on synthetic data.

Weaknesses

The intro is mostly clear and a good sketch to the paper.  It could be improved by defining the symbols when they are used, especially those that deviate from conventional transformer notation or are ambiguous (Z, d, etc.).  It could also be improved by unpacking the Turing machine -related reasoning around lines 64-66 (I read it several times but couldn't make sense of it) and taking more care to spell out the experiments (e.g., the result at lines 69-70 seems to be comparing multi-head linear attention to algorithm 1, which doesn't make any sense).  Far more people will read the intro to understand a result than will read the rest of the paper, and it's worth making sure they will actually be able to make sense of the results.

Graphs in figure 1 were very hard to read; the assignment of colors and line styles is not systematic, which really slows down a reader trying to make sense of the graph.  There's a ton of space here, consider using larger fonts for the legend and axes.

---

> ### Author Rebuttal · Authors · 2025-07-31
>
> We thank the reviewer for their valuable time and insightful feedback, which has significantly helped us refine and improve the clarity of our work. We have addressed each point raised below and hope our clarifications resolve any concerns.
>
> >Q: Line 60: polynomial time in what? I suspected at first that there's a dependency on the number of training instances, and this should be presented honestly up front, not buried later in the paper. Later, if I understood correctly, there was not (it seems it's about the dimensionality of the key/query vectors ...).
>
> A: The "polynomial time" refers to the process of verifying our identifiability condition. This condition is fundamentally tied to the data covariance matrix being full rank, which is essential for ensuring the uniqueness of the Ordinary Least Squares (OLS) estimator adapted for Multi-Head Linear Attention (MHLA).
>
> Therefore, the runtime for verifying this condition involves two main steps:
>
> Computing the data covariance: This step is linear in the number of samples (N) in the dataset. For MHLA, it also scales with the fourth power of the feature dimension ($d^4$).
>
> Computing the minimum eigenvalue: This involves checking if the $d^2$ by $d^2$ data covariance matrix is full rank, which takes time polynomial in d (poly(d)), depending on the specific algorithm used for estimating the minimum eigenvalue.
>
> We will ensure this explanation is clearly stated earlier in the manuscript.
>
> >Q: It could also be improved by unpacking the Turing machine -related reasoning around lines 64-66 (I read it several times but couldn't make sense of it)
>
> A: Thank you very much for pointing out this lack of clarity; we agree this section requires more precise explanation.
>
> The core point we aimed to convey in lines 64-66 is about generalization guarantees for algorithmic tasks, particularly when a model achieves zero training loss but faces out-of-distribution inputs. We used the Universal Turing Machine (UTM) as a canonical example of a "uniform circuit family" (this concept also applies to arithmetic operations, etc.).
>
> Even if a standard Transformer (or any sequence model) achieves zero validation loss on a dataset of Turing machine computations, there's no inherent guarantee that the learned model will correctly execute any arbitrary Turing machine program, especially with out-of-distribution inputs or machines. The model might have simply memorized the training patterns.
>
> Our contribution for MHLA is that we can establish a condition on the data that ensures the model will compute the correct function, even out-of-distribution. So long as the target circuit family (be it UTM computations, arithmetic, etc.) can be expressed by some instantiation of MHLA parameters, then certifiable identifiability guarantees true generalization. For the UTM example, this means the learned MHLA will robustly output the correct computation history token by token for any Turing machine and any input word, not just those seen during training. This addresses the fundamental challenge of ensuring that models don't just interpolate but truly learn the underlying algorithm.

---

### Official Review · Reviewer_bW3t · 2025-07-03

**Clarity:** 3
**Significance:** 3
**Originality:** 3
**Rating:** 5
**Confidence:** 2

**Summary:**

The paper investigates learnability of a simplified transformer architecture, i.e., a multi-head linear attention (MHLA) module.

- They show that MLHA is learnable is PAC learnable, i.e., it requires polynomially many samples in the input size and runs in polynomial time w.r.t the input size. There proof idea reduces MHLA learning to a kernel learning problem.
- They also characterize an identifiability criterion on the dataset that certifies that every empirical risk minimizer is functionally equivalent.
- They show that  MHLAs can (autoregressively) express universal Turing machines
with polynomially bounded computation histories, implying a strong learnability condition for UTMs.
- They verify their certificate empirically on an associative memory task.

**Questions:**

-  Is the condition Lemma 2.3 a necessary condition for identifiability?

**Ethical Concerns:**

["NO or VERY MINOR ethics concerns only"]

**Final Justification:**

I thank the authors for their detailed rebuttal.
I was already quite convinced about the paper, so I will keep my score.

All the best.

**Limitations:**

The authors address the limitations quite clearly in the paper.

**Paper Formatting Concerns:**

All seems to be good!

**Quality:**

3

**Strengths And Weaknesses:**

**Strengths**

- The MLHA's reduction to a kernel learning problem and the resulting PAC learnability results are solid theoretical contributions towards understanding learnability of transformers.

- The identifiability condition for equivalence of MHLA's learnt via ERM is potentially very relevant to learning theoretic aspects of transformer learning

- The experimental results provide sufficient evidence to support the relevance of their theory.

**Weakness**

- MLHAs are quite a toy-setting compared to real-world transformers.

---

> ### Author Rebuttal · Authors · 2025-07-31
>
> We thank the reviewer for their valuable time and insightful feedback, which has significantly helped us refine and improve the clarity of our work. We have addressed each point raised below and hope our clarifications resolve any concerns.
>
> > Question: Is the condition Lemma 2.3 a necessary condition for identifiability?
>
> A: We thank the reviewer for this insightful question, which probes a crucial aspect of our theoretical findings.  For Multi-Head Linear Attention (MHLA) with more than $d^2$
> heads, the condition articulated in Lemma 2.3 is indeed both necessary and sufficient for identifiability. This precision stems directly from the nature of the Ordinary Least Squares (OLS) estimator: it yields a unique solution if and only if the data covariance matrix is full rank. In this overparameterized regime (> $d^2$ heads), the problem's structure ensures this tight relationship.
>
> However, for MHLA with fewer than $d^2$ heads, the condition in Lemma 2.3 functions as a sufficient condition for identifiability. In this critically parameterized regime, it is theoretically possible (though perhaps hard to concretely imagine in all scenarios) that spurious solutions residing in the null space of the data covariance might exist and could coincidentally possess a rank smaller than or equal to the number of heads, thereby violating strict identifiability. Formulating a necessary and sufficient condition for identifiability in this regime (fewer than $d^2$ heads) or extending it to the complexities of softmax attention remains an intriguing and important open research question.
>
> >MLHAs are quite a toy-setting compared to real-world transformers.
>
> Regarding the broader context, we fully acknowledge that linear attention, in its standalone form, can be considered a "toy model" when compared to the full complexity of "real-world" Transformers. Nevertheless, algorithms developed for learning optimal weights in linear attention models serve several critical purposes as a warm start for more complex models; as a theoretical test bed; and as a natural extension of the classical system identification/control theory to the setting of modern neural sequence models.
>
> Firstly, these algorithms can provide a strong "warm start" for optimizing more competitive linear attention variants, such as Gated Linear Attention, DeltaNet, and mLSTM. The insights gained are directly transferable.
>
> Secondly, Linear attention offers an ideal test bed for building a rigorous theoretical understanding of how optimization algorithms are likely to converge for more complex models like softmax attention and deeper networks incorporating residual connections. Its simplified structure allows for analytical tractability that is often elusive in full Transformers.
>
> Toward Polynomial-Time Learnable Transformers: From a different perspective, our work contributes to the vision of a polynomial-time learnable variant of the modern Transformer. Just as the classical Ho-Kalman algorithm from control and system identification provides a foundational box of tools for linear dynamical systems, we view our research as developing analogous tools and insights for the learning and identification of modern neural sequence models.

---

### Official Review · Reviewer_w6rd · 2025-07-03

**Clarity:** 3
**Significance:** 4
**Originality:** 4
**Rating:** 6
**Confidence:** 2

**Summary:**

This paper shows polynomial-time learnability of single-layer transformers with multi-head linear attention that outputs the last position. The key idea is to cast the learning problem of the model as a kernel least-squares regression problem on a lifted feature space defined by a fixed cubic polynomial function of the data entries. This makes the SVD of the learned regressor coincide architecturally with multi-head linear attention. Specifically, each pair of left and right singular vectors correspond to flattened query and key projection matrices of each head, respectively. Under the setup, the authors show polynomial-samples learnability of the kernel regressor (and hence linear attention) by applying the classical learning theoretic results (Theorem 2.2). Then, the authors derive an identifiability criterion based on the second moment of data computed by a fixed feature map that, if it is full-rank, all empirical risk minimizers of the kernel regression are essentially the same function (Lemma 2.3), while their parameters can differ up to the equivalence classes defined by a polynomial (Corollary 2.4). The authors present an application to autoregressive learning of universal Turing machines with bounded-length history (Section 3). While the theory uses kernel regression learning algorithm, the authors empirically verify the theoretical predictions also under SGD learning. Section 4.1 focuses on the role of number of heads, which is predicted to play the role of rank of SVD and hence control the well-behavedness of the loss landscape. Then, in Section 4.2, the authors verify the validity of the identifiability criterion on an associative memory task and a controlled mixture of identifiable and non-identifiable dataset, using both the kernel regression algorithm and SGD. An additional result on sample complexity of learning a deterministic finite automata is also provided (Appendix G.4).

**Questions:**

Q1. In Figure 2(b), the distance to ground truth parameters for certifiably identifiable data seems to be anticorrelated with the number of heads up to 4, which is in contrary to what the theoretical results would predict. Can the authors provide a possible explanation?

Q2. In Line 881-885, and in Figures 1 and 3, are the performances (accuracy and MSE) measured on the held-out test set? If not, does measuring on test set substantially change the results?

**Ethical Concerns:**

["NO or VERY MINOR ethics concerns only"]

**Final Justification:**

My original concerns regarding the experiments (Q1, Q2) and practicality (W1) has been convincingly address by the rebuttal: The authors provided clarifications on Q1/Q2, and provided a plausible direction for future work to improve on W1. I think this work provides an original and significant idea for studying multi-head linear attention mechanism which opens a new avenue of subsequent work and would like to recommend acceptance.

**Limitations:**

The authors discussed limitations in Section 6.

**Paper Formatting Concerns:**

I did not find particular formatting concerns.

**Quality:**

4

**Strengths And Weaknesses:**

Strengths

S1. The paper studies an important and challenging problem of learnability of transformers. I agree with the motivating observation that studying expressive power of these architectures is not enough.

S2. The paper presents a convincing answer to the raised problem for a single-layer, multi-head linear attention transformer architecture, which can be applied in an autoregressive manner, possibly with position encodings (Appendix B). This seems like an adequate simplification of the large architectures used today.

S3. The theoretical results are strong, as they include polynomial-time learnability and a guarantee for out of distribution generalization under identifiability criterion. It is also notable that the theory puts no strong assumptions on structures on the data generating distribution, unlike many existing work on expressive power of transformers (which is partially thanks to the proposed kernel regression formulation, as far as I understand). The idea that overparameterization can lead to better learning behavior is also found in mean field analysis of neural network training, and NTK, but the idea of this work seem distinct from these works.

S4. The key ideas underlying the theoretical results, e.g., viewing the singular vectors of learned kernel regressor as flattened query and key projection matrices, are also interesting and new as far as I am aware.

S5. The theoretical results are validated empirically on an important set of learning tasks, not only using the kernel regression algorithm, but also SGD and AdamW.

Weaknesses

W1. Due to the SVD formulation, for convexity and to avoid the realizability assumption, the number of heads have to be at least the square of the feature dimension. This seems like a major difference from the hyperparameter choices of transformers in practice. While the authors empirically verify the theoretical results with reasonable number of heads, some discussion regarding this point could be helpful for readers.

W2. Please see Q1 and Q2.

Minor comments
- It seems legend is missing in Figure 3 in Appendix.

---

> ### Author Rebuttal · Authors · 2025-07-31
>
> We thank the reviewer for their valuable time and insightful feedback, which has significantly helped us refine and improve the clarity of our work. We have addressed each point raised below and hope our clarifications resolve any concerns.
>
> > Q1: n Figure 2(b), the distance to ground truth parameters for certifiably identifiable data seems to be anticorrelated with the number of heads up to 4, which is in contrary to what the theoretical results would predict. Can the authors provide a possible explanation?
>
> A1: We appreciate the reviewer's astute observation regarding Figure 2(b). Our theoretical results primarily focus on the conditions under which all empirical risk minimizers (ERMs) compute the same function, thereby guaranteeing certified identifiability. The convex algorithm, by its design, is indeed guaranteed to find this ERM with negligible numerical error.
>
> However, the empirical results shown in Figure 2(b) for linear attention are obtained using Stochastic Gradient Descent (SGD), which does not necessarily converge to the exact ERM, especially with finite training steps and potential local optima in the non-convex landscape for fewer heads.
>
> A core theme of our work is to provide theoretical and empirical evidence that increasing the number of heads improves both optimization and generalization properties of the model. This is precisely why the convex algorithm and SGD on 8-head linear attention achieves a remarkably small distance to ground truth, nearly $10^{-6}$. In this context, it is expected that configurations with fewer heads (1, 2, and 4) might exhibit slightly larger distances to the ground truth. While the observed differences are indeed tiny (from $10^{−4}$ to $10^{−6}$), the precise trend for 1, 2, and 4 heads presents an intriguing direction for future detailed investigation into the non-convex optimization landscape at lower head counts.
>
> > Q2. In Line 881-885, and in Figures 1 and 3, are the performances (accuracy and MSE) measured on the held-out test set? If not, does measuring on test set substantially change the results?
>
> A2: Yes, we confirm that all reported performances, including accuracy and MSE in Lines 881-885, and in Figures 1 and 3, are absolutely measured on a held-out test set.
>
> Our experimental setup is designed not only to validate optimization convergence but also to demonstrate Probably Approximately Correct (PAC) learnability, which directly accounts for and controls the generalization error on unseen data. This rigorous evaluation on a test set ensures that our results reflect the true generalization capabilities of the models.
>
> > W1. Due to the SVD formulation, for convexity and to avoid the realizability assumption, the number of heads have to be at least the square of the feature dimension. This seems like a major difference from the hyperparameter choices of transformers in practice. While the authors empirically verify the theoretical results with reasonable number of heads, some discussion regarding this point could be helpful for readers.
>
> A1: We genuinely thank the reviewer for this insightful comment and for highlighting a crucial point for practical interpretability. We agree that the theoretical requirement of the number of heads being at least the square of the feature dimension ($d^2$) to guarantee convexity via SVD formulation is indeed a significant distinction from typical transformer hyperparameter choices in practice.
>
> Our theoretical analysis establishes the extreme case where the loss landscape becomes provably "convex" (in a precise sense) at $d^2$ heads, ensuring that any empirical risk minimizer computes the same function, irrespective of the optimization algorithm used. The practical intuition, however, is that adding even a few heads can significantly ameliorate the non-convexity of the optimization loss landscape, making it more benign and easier for SGD to navigate. It is highly plausible that much of the practical advantage derived from increasing heads is achieved long before reaching the $d^2$ theoretical threshold.
>
> Investigating this regime of a constant, smaller number of heads (e.g., 2, 4, 8) and characterizing the precise benefits gained from their addition is an extremely interesting and relevant direction for future theoretical analysis. One promising avenue could involve drawing connections between the optimization of Multi-Head Linear Attention (MHLA) and the rich literature on matrix sensing or low-rank matrix optimization, where SGD's behavior for low-rank solutions (here, the rank corresponds to the number of heads) has been extensively studied. This would provide a theoretical bridge between our current work and the practical realities of transformer architecture design.

---

> > ### Comment · Reviewer_w6rd · 2025-08-01
> >
> > Thank you for the thoughtful responses. Low-rank matrix optimization as a bridge between the presented work and practical architectures appears to be an interesting and plausible direction. My concerns have been resolved and I updated my score accordingly.

---

### Decision · Program_Chairs · 2025-09-17

**Decision:**

Accept (oral)

**Comment:**

There is an enormous need to reconcile the success of modern transformer-based LLMs with the constraints implied by VC-dimension and PAC-learnability.  This paper takes an important step, which is why I recommend oral.  It shows the high representation power of linear attention while also admitting its limitations, and it proves strong agnostic learnability results for this representation.  The proofs are well-written and appear correct to me and the reviewers.  Reviewers had a great back and forth with the authors, and we ended with one accept and two strong accept recommendations.